# A genetic compensatory mechanism regulated by *Jun* and *Mef2d* modulates the expression of distinct class IIa *Hdacs* to ensure peripheral nerve myelination and repair

**Sergio Velasco-Aviles[1,2†], Nikiben Patel[1,2†], Angeles Casillas-Bajo[1,2], Laura Frutos-Rincón[1,3], Enrique Velasco[1,3], Juana Gallar[1,2,3,4], Peter Arthur-Farraj[5], Jose A Gomez-Sanchez[1*], Hugo Cabedo[1,2*]**

[1]Instituto de Neurociencias de Alicante UMH-CSIC, Alicante, Spain; [2]Instituto de Investigación Sanitaria y Biomédica de Alicante (ISABIAL), Alicante, Spain; [3]The European University of Brain and Technology-NeurotechEU, Alicante, Spain; [4]RICORS en enfermedades inflamatorias, Sant Joan d'Alacant, Spain; [5]John Van Geest Centre for Brain Repair, Department of Clinical Neurosciences, University of Cambridge, Cambridge, United Kingdom

**\*For correspondence:**
j.gomez@umh.es (JAG-S);
hugo.cabedo@umh.es (HC)

†These authors contributed equally to this work

**Competing interest:** The authors declare that no competing interests exist.

**Abstract** The class IIa histone deacetylases (HDACs) have pivotal roles in the development of different tissues. Of this family, Schwann cells express *Hdac4*, *5*, and *7* but not *Hdac9*. Here, we show that a transcription factor regulated genetic compensatory mechanism within this family of proteins, blocks negative regulators of myelination ensuring peripheral nerve developmental myelination and remyelination after injury. Thus, when *Hdac4* and *5* are knocked-out from Schwann cells in mice, a JUN-dependent mechanism induces the compensatory overexpression of *Hdac7* permitting, although with a delay, the formation of the myelin sheath. When *Hdac4*, *5*, and *7* are simultaneously removed, the myocyte-specific enhancer-factor d (MEF2D) binds to the promoter and induces the de novo expression of *Hdac9*, and although several melanocytic lineage genes are misexpressed and Remak bundle structure is disrupted, myelination proceeds after a long delay. Thus, our data unveil a finely tuned compensatory mechanism within the class IIa *Hdac* family, coordinated by distinct transcription factors, that guarantees the ability of Schwann cells to myelinate during development and remyelinate after nerve injury.

## Editor's evaluation

Analyzing Schwann cells which make myelin in the mammalian peripheral nervous system, the authors unravel how transcription factors can functionally substitute each other in the development of single/double/ and triple mutant mice. Functional redundancy and compensation is seen in many developmental systems, but has rarely been studied at that level of detail. The paper is thus of interest also to scientists beyond the field of glial cell biology.

## Introduction

During the postnatal development of the peripheral nervous system (PNS), immature Schwann cells ensheath large caliber axons of sensory and motor neurons and differentiate, forming myelin, a highly

specialized plasma membrane that increases nerve impulse velocity by allowing saltatory conduction (*Jessen and Mirsky, 2005*). Immature Schwann cells downregulate the transcription factor *Jun* (which negatively regulates myelination) and upregulate the expression of transcriptional regulators of myelination such as *Krox-20* and *Yy1* (*Fazal et al., 2017*; *Monk et al., 2015*; *Parkinson et al., 2008*). *Jun* is strongly reexpressed after nerve injury enabling trans-differentiation of Schwann cells into a repair phenotype that promotes axon regeneration and functional nerve repair (*Arthur-Farraj et al., 2012*; *Gomez-Sanchez et al., 2015*). After axon regeneration Schwann cells reestablish contact with them and downregulate *Jun*. This allows reexpression of *Krox-20* and the consequent reactivation of a gene expression program aimed at remyelination of axons and reestablishment of nerve function (*Stassart and Woodhoo, 2021*). Activation of Gpr126, a G-protein-coupled receptor that increases intracellular levels of cAMP, is required for Schwann cell myelination and remyelination (*Monk et al., 2009*; *Monk et al., 2011*). We have recently shown that the prodifferentiating activity of cAMP is in part mediated by its ability to shuttle HDAC4 into the nucleus of Schwann cells (*Gomis-Coloma et al., 2018*). Nuclear HDAC4 recruits the complex NcoR1/HDAC3 and deacetylates histone three on the promoter of *Jun*, repressing its expression. At the same time HDAC4 promotes *Krox-20* expression and activation of the myelination program (*Velasco-Aviles et al., 2018*). In vivo, *Hdac5* is able to partially compensate for the loss of *Hdac4* expression in Schwann cells and only the removal of both *Hdac4* and *Hdac5* from Schwann cells leads to an obvious myelination delay. Surprisingly by postnatal day 8, myelination in *Hdac4/5* double knockout mice proceeds at the same pace as in wild-type nerves, suggesting that there is an additional compensatory mechanism permitting nerve myelination (*Gomis-Coloma et al., 2018*). Here, we show that the in vivo elimination of *Hdac4* and *Hdac5* from Schwann cells induces the overexpression of *Hdac7* through a mechanism mediated by the transcription factor JUN. Notably, the removal of *Hdac7* from Schwann cells in the absence of *Hdac4* and *Hdac5* produces a much longer delay in myelin development. This demonstrates that overexpressed *Hdac7* can partially compensate for the absence of both *Hdac4* and *Hdac5* in myelinating Schwann cells. Interestingly, nonmyelin-forming Schwann cells in these triple knock-outs (KOs) misexpress melanocytic lineage genes and fail to properly segregate small caliber axons in the Remak bundles. We show that genetic compensation also plays a pivotal role during remyelination after nerve injury. Thus, and akin to what happens during development, remyelination is delayed when *Hdac4* and *Hdac5* are removed from Schwann cells. This delay is longer when *Hdac7* is also removed, which has a profound impact on nerve impulse conduction during nerve regeneration. Importantly, remyelination in the *Hdac4/5/7* triple KO also catches up, supporting the idea that an additional mechanism compensates for the absence of class IIa *Hdacs*. Strikingly, *Hdac9*, the only class IIa *Hdac* that is not normally expressed by Schwann cells, is de novo expressed in the nerves of the *Hdac4/5/7* triple KO mice, induced by the transcription factor MEF2D. These genetic compensatory mechanisms, centering around transcription factors, allow Schwann cells to retain a class IIa *Hdac* gene dosage high enough to permit eventual myelination during development and remyelination after injury.

## Results

### Upregulation of *Hdac7* permits developmental myelination in the absence of *Hdac4* and *Hdac5*

We have previously shown that *Hdac4* and *Hdac5* redundantly contribute to activate the myelin transcriptional program in Schwann cells in vivo. However, although during postnatal development *Jun* levels remain high in the PNS of the *Hdac4/5* double conditional knock out mice (*Mpz-Cre*$^{+/-}$; *Hdac4*$^{flx/flx}$;*Hdac5*$^{-/-}$, hereafter called dKO), myelination proceeds normally after P8 and adult nerves are morphologically and functionally indistinguishable from those of wild-type mice (*Gomis-Coloma et al., 2018*). In muscle development class IIa Hdacs can compensate for each other (*Potthoff et al., 2007b*). In addition to *Hdac4* and *Hdac5*, Schwann cells also express *Hdac7* (*Gomis-Coloma et al., 2018*). To test if it can functionally compensate for the absence of *Hdac4* and *Hdac5*, we measured the expression levels of *Hdac7* in the nerves of dKO. As shown in *Figure 1A*, the expression of *Hdac7* was substantially induced in the sciatic nerve of the dKO mice at P60 (325.1 ± 48.1%; p = 0.0034, *n* = 4), while *Hdac9* expression remained residual. This is specific for the dKO, as minor or no changes at all were found in the single KOs (*Figure 1—figure supplement 1A, B*). Importantly, *Hdac7* overexpression can be detected early during development (*Figure 1—figure supplement 1C*). These results

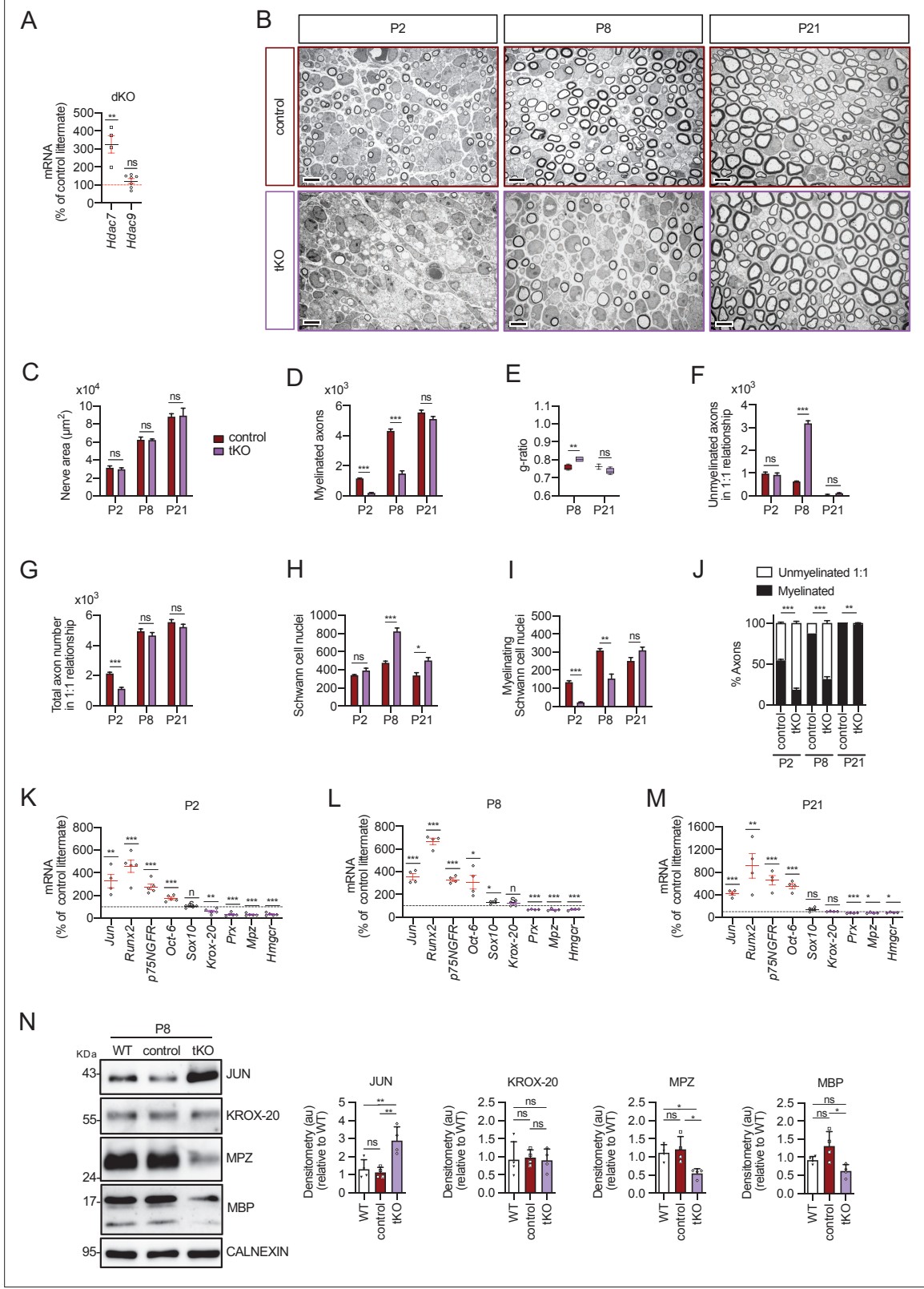

**Figure 1.** Myelin development is notably delayed in the tKO mice. (**A**) A 325.1 ± 48.1% (p = 0.0034) increase in the amount of mRNA for *Hdac7* was found in the dKO nerves. No changes in the expression of HDAC9 were found. RT-qPCR with mouse-specific primers for *Hdac7* was performed and normalized to 18S rRNA. The scatter plot, which include also the mean ± standard error (SE), shows the fold change in mRNA normalized to control littermates. Four to eight mice per genotype were used. Data were analyzed with the unpaired *t*-test. (**B**) Representative transmission TEM mages of P2,

*Figure 1 continued on next page*

*Figure 1 continued*

P8, and P21 sciatic nerves of tKO mice (*Mpz-Cre$^{+/-}$;Hdac4$^{flx/flx}$;Hdac5$^{-/-}$;Hdac7$^{flx/flx}$*) and the control (*Mpz-Cre$^{-/-}$;Hdac4$^{flx/flx}$; Hdac5$^{-/-}$;Hdac7$^{flx/flx}$*) littermates. Scale bar: 5 µm. (**C**) No statistically significant differences were observed between the area of the tKO nerves and control littermates (P2: p = 0.5234; P8: p = 0.9279; P21: p = 0.9009). (**D**) The number of myelinated axons is notably decreased at P2 (208 ± 24 in tKO versus 1.160 ± 29 in controls; p ≤ 0.0001) and P8 (1.487 ± 179 in tKO versus 4.235 ± 129 in controls; p ≤ 0.0001). (**E**) g ratio was increased at P8 (0.80 ± 0.01 in the tKO versus 0.76 ± 0.01 in control; p = 0.0045). (**F**) The number of unmyelinated axons in a 1:1 relationship with Schwann cells was notably increased at P8 (3.187 ± 111 in the tKO versus 628 ± 21 in controls; p ≤ 0.0001). (**G**) The total number of sorted axons in a 1:1 relationship with Schwann cells is decreased at P2 (1.128 ± 90 in the tKO versus 2.131 ± 95 in the control; p = 0.0007). (**H**) The total number of Schwann cells (counted as nuclei) is increased at P8 (823 ± 37 in the tKO versus 476 ± 20 in controls; p ≤ 0.0001) and at P21 (503 ± 31 in the tKO versus 337 ± 32 in controls; p ≤ 0.0152). (**I**) In contrast, the number of myelinating Schwann cells is decreased at P2 (22 ± 1 in the tKO versus 134 ± 8 in controls; p ≤ 0.0001) and at P8 (153 ± 25 in the tKO versus 309 ± 11 in controls; p = 0.0013). (**J**) The percentage of myelinated axons is decreased at P2 (18.5 ± 3.7% in the tKO versus 54.6 ± 1.1% in controls; p ≤ 0.0001), P8 (31.6 ± 2.9% in the tKO versus 54.6 ± 1.1% in controls; p ≤ 0.0001) and, although much less, at P21 (97.9 ± 0.4% in the tKO versus 99.9 ± 0.0% in controls; p = 0.0135). For these experiments, three to four animals per genotype were used; unpaired *t*-test was applied for statistical analysis. (**K**) Markers of nonmyelin-forming Schwann cells are upregulated whereas those of myelin-forming Schwan cells are downregulated in the tKO. P2, sciatic nerves were removed and total RNA extracted. RT-qPCR with mouse-specific primers for the indicated genes was performed and normalized to 18S rRNA. Graph shows the percentage of mRNA for each gene in the tKO normalized to the control littermates. A scatter plot is shown with the results obtained, which include also the mean ± SE. (**L**) The same for P8. (**M**) The same for P21. For these experiments, four to five mice per genotype and age were used. Data were analyzed with the unpaired *t*-test. (**N**) A representative WB of protein extracts from tKO, control, and wild-type P8 nerves is shown. In the quantification, JUN protein increased in the tKO (2.88 ± 0.19 in the tKO versus 1.12 ± 0.071 in the control nerve; p = 0.004). Mpz protein was found decreased (0.55 ± 0.03 in the tKO versus 1.21 ± 0.09 in the control nerve; p = 0.0115) as was Mbp (0.62 ± 0.045 in the tKO versus 1.31 ± 0.100 in the control nerve; p = 0.012). We could not find changes in KROX-20. Densitometric analysis was done for seven to nine WB from the same number of mice and normalized to the WT. Data were analyzed with the one-way analysis of variance (ANOVA) Tukey's test (*p < 0.05; **p < 0.01; ***p < 0.001; ns: no significant). See source data file one online (graphs source data) for more details.

The online version of this article includes the following figure supplement(s) for figure 1:

**Figure supplement 1.** Class IIa HDAC gene expression and removal from Schwann cells.

**Figure supplement 2.** Myelin development in the cKO4 mice sciatic nerves.

**Figure supplement 3.** Myelin development in the KO5 mice sciatic nerves.

**Figure supplement 4.** Myelin development in the cKO7 mice sciatic nerves.

**Figure supplement 5.** Myelin development in the dKO mice sciatic nerves.

suggest that the simultaneous elimination of *Hdac4* and *Hdac5* from Schwann cells activates a mechanism aimed to compensate for the drop in the gene dose of *class IIa Hdacs* that upregulates threefold the expression of *Hdac7*. To test whether *Hdac7* can functionally compensate to allow myelination in the absence of *Hdac4/5*, we generated a *Hdac4/5/7* triple Schwann cell-specific conditional KO (genotype *Mpz-Cre$^{+/-}$; Hdac4$^{flx/flx}$;Hdac5$^{-/-}$; Hdac7$^{flx/flx}$*, hereafter called tKO; ***Figure 1—figure supplement 1D—F***). To study myelin development in these mice we evaluated a number of morphological parameters of sciatic nerves at P2, P8, and P21 using transmission electron microscopy (TEM) images. We also quantified the mRNA and protein levels for a number of negative and positive regulators of myelination. We previously separately analyzed Schwann cell gene expression in sciatic nerves of *Hdac4* conditional KO mice (genotype *Mpz-Cre$^{+/-}$; Hdac4$^{flx/flx}$*, referred to as cKO4) mice and global *Hdac5* KO mice (referred to as KO5) (***Gomis-Coloma et al., 2018***). As additional controls, here we also performed a detailed morphological analysis of developing nerves in cKO4, KO5, dKO, and *Hdac7* Schwann cell-specific conditional KO mice (*Mpz-Cre$^{+/-}$; Hdac7$^{flx/flx}$* referred to as cKO7) (***Figure 1— figure supplements 2–5***). As is shown in ***Figure 1—figure supplements 2–4***, morphological quantification showed that within the single mutants, only cKO4 showed a subtle, but consistent, delay in myelin development. In line with our previous results (***Gomis-Coloma et al., 2018***), the simultaneous elimination of *Hdac4* and *Hdac5* from Schwann cells produced a greater decrease in the percentage of myelinated axons at P2 that was almost normalized by P8 (***Figure 1—figure supplement 5***). Strikingly, the simultaneous elimination of *Hdac4*, *5*, and *7* from Schwann cells produced a much more pronounced delay in myelin development (***Figure 1B–J***). Interestingly, expression of the negative regulators of myelination (including *Jun*) was notably increased from P2 to P21, which can explain the delay in the expression of myelin genes (***Figure 1K–M***) and in morphological parameters of myelin development in the tKO mice. Thus, our data demonstrate that *Hdac7* upregulation can compensate for the absence of *Hdac4* and *Hdac5* allowing myelination to proceed, although with some delay. Interestingly, and although the coordinated removal of *Hdac4*, *Hdac5*, and *Hdac7* produces a long

delay in myelination, myelin is finally formed and adult tKO nerves show almost normal myelination parameters (*Figure 2A–D*).

## Defects in Remak Schwann cell differentiation in the tKO

Despite PNS myelination looks normal in the adult (p60) tKO mice, we found the Remak bundles profoundly altered in these nerves, with many axons not properly segregated (*Figure 2E*). Thus, there is a significant increase in the number of pockets with two to five axons and, although it is very rare to find pockets with more than five axons in the control (2.3%), an important number of axons are grouped together in packs of more than five in the tKO (16.5%), with some of them being in pockets of more than 30 axons. These defects are specific of the tKO, as no major changes were observed in the single neither the dKO nerves (*Figure 2—figure supplement 1A*).

Whole genome-wide transcriptome analysis showed 654 upregulated and 616 downregulated genes in the nerves of adult tKO (*Figure 2F* and source data file two online [RNA-seq source data]). Volcano plot shows that genes tended toward being more strongly upregulated than downregulated. Surprisingly, the most robustly upregulated gene is the tyrosinase-related protein one encoding gene (*Tyrp1*; Log FC = 6.03; FDR = 0), a melanocyte lineage-specific gene (*Figure 2G, H*). Additionally, the melanoma cell adhesion molecule *Mcam* and *Ngfr* genes are also highly induced in the sciatic nerves of the tKO. RT-qPCR confirmed the strong induction of *Tyrp1* and *Mcam* in the sciatic nerves of the tKO (P60), but not in the single KOs neither dKO nerves (*Figure 3A, B*). Interestingly we also found increased the expression of Microphthalmia-associated transcription factor (*Mitf*) and the Endothelin B receptor (*Ednrb*), two other genes of the melanocytic lineage (*Figure 3C, D*). Importantly, the expression of all these genes increased from early in postnatal development (*Figure 2—figure supplement 1B, C*). Western blot analysis (*Figure 3E, F*) and confocal microscopy confirmed these findings and showed that the misexpression of melanocytic lineage markers is confined to the nonmyelin-forming Schwann cells of the Remak bundles (*Figure 3G–K*). Thus, our data suggest that class IIa HDACs are necessary to allow Schwann cell precursors (SCPs) to differentiate into Remak Schwann cells and properly segregate small size axons.

## Remyelination kinetics after nerve injury depends on class IIa *Hdac* gene dose

The molecular mechanisms of Schwann cell remyelination share similarities to myelination during development, however there are also notable differences (*Stassart and Woodhoo, 2021*). Given the role of class IIa HDACs in myelination during development we asked whether they are also involved in remyelination after nerve injury. To address this, we first performed crush experiments in the sciatic nerves of 8-week-old cKO4 mice. As controls, we used *Mpz-Cre$^{-/-}$;Hdac4$^{flx/flx}$* littermates (*Figure 4—figure supplement 1A—K*). At 10 days postinjury (dpi), we found a small decrease in the percentage of myelinated axons, with an increase in the number of unmyelinated axons with a diameter >1.5 µm in a 1:1 relationship with the Schwann cells (*Figure 4—figure supplement 1F—K*). A small increase in the number of Schwann cell nuclei was also found at 20 and 30 dpi (*Figure 4—figure supplement 1I*). Also, subtle but significant changes in myelin protein gene expression were found in the cKO4 nerves (*Figure 4—figure supplement 1L–N*). Notably, myelin clearance was normal ruling out this as the cause of remyelination delay (*Figure 4—figure supplement 1O*). Thus, *Hdac4* removal has as small impact on remyelination that is compensated for after 10 dpi.

Morphological analysis of remyelination in KO5 mice and wild-type littermates (*Hdac5$^{+/+}$*) showed no differences (*Figure 4—figure supplement 2*). Also, no notable differences in myelin protein gene expression (*Figure 4—figure supplement 1L–N* and *Figure 4—figure supplement 2L–M*) nor myelin clearance were found between both genotypes (*Figure 4—figure supplement 1O* and *Figure 4—figure supplement 2N*).

To explore whether there is also genetic compensation within class IIa *Hdacs* in Schwann cells after injury, we analyzed remyelination in the dKO crushed nerve. In this case, the percentage of myelinated axons at 10 days after crush (10 dpi) was notably decreased (15.5 ± 2.3% in the dKO versus 60.4 ± 4.8% in the control; p ≤ 0.0001) (*Figure 4A, K*). Total axon counts were similar between genotypes suggesting that this difference was not due to an axon regeneration defect (*Figure 4G*). At 20 dpi the difference between both genotypes was reduced and normalized at 30 dpi. A notable increase in the number of unmyelinated axons with a diameter >1.5 µm in a 1:1 relationship with the Schwann

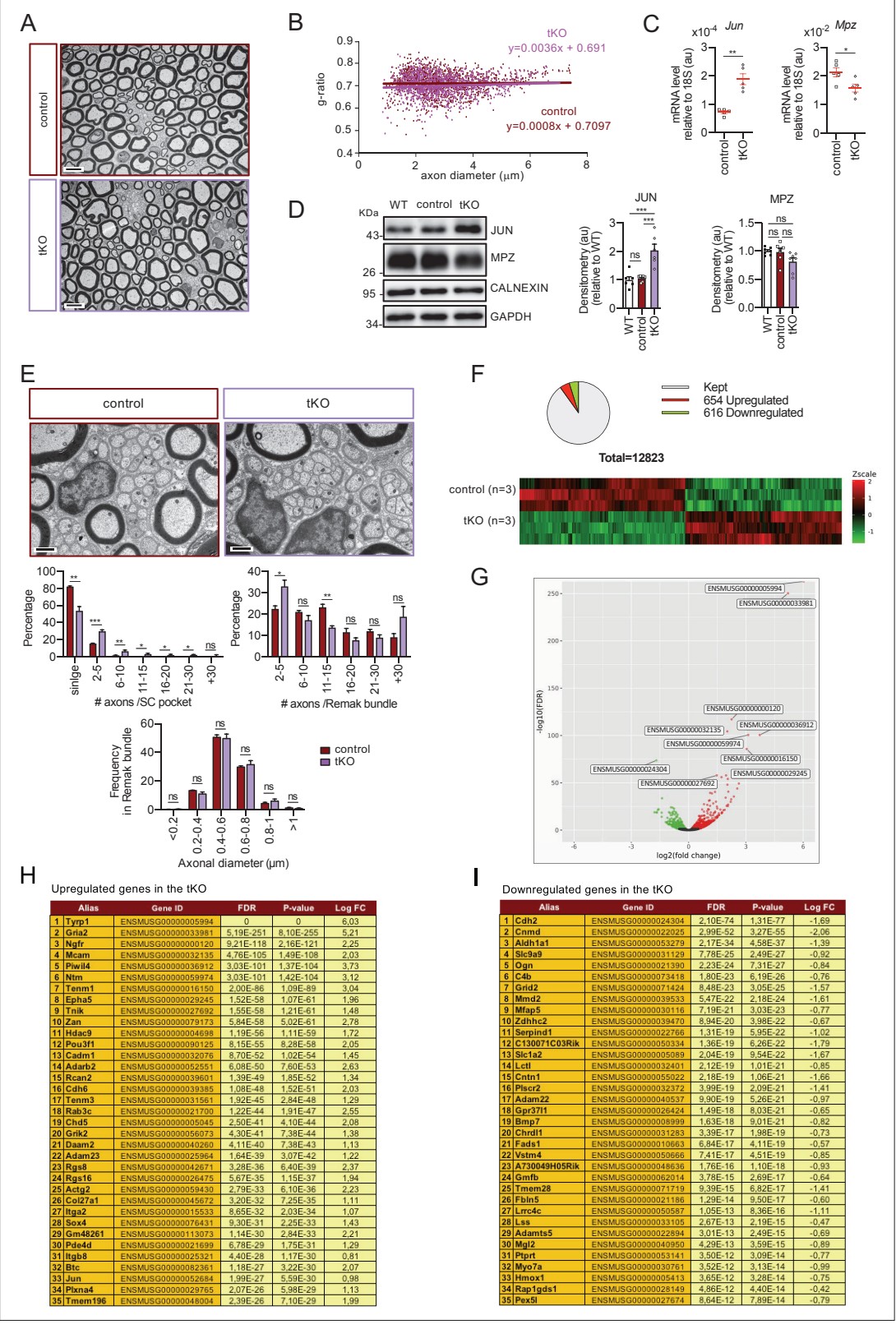

**Figure 2.** Characterization of the tKO. (**A**) Representative transmission TEM image of the sciatic nerve of and adult (P60) tKO mouse and a control littermate. Scale bar: 5 μm. (**B**) Scatter plot of *g* ratio versus axon diameter. 1100 axons of 4 different mice per genotype were used. No changes in *g* ratio were detected. (**C**) mRNA for *Jun* remains increased in the tKO by 2.6-fold (1.88 ± 0.19 × $10^{-4}$ au in the tKO versus 0.72 ± 0.05 × $10^{-4}$ au in controls; p = 0.003) whereas *Mpz* was slightly decreased (1.57 ± 0.13 × $10^{-2}$ au in the tKO versus 2.13 ± 0.05 × $10^{-2}$ au in controls; p = 0.027). RT-qPCR

*Figure 2 continued on next page*

*Figure 2 continued*

with mouse-specific primers for the indicated genes was performed. Graph shows a scatter plot for the ΔCt (which include also the mean ± standard error [SE]) of the gene normalized to the housekeeping 18S. Five mice per genotype and age were used. Data were analyzed with the unpaired *t*-test. (**D**) JUN and MPZ protein levels. A representative Western blot of protein extracts from wild-type (C57BL/6), control and tKO sciatic nerves is shown. The densitometric analysis of six to seven different experiments normalized to WT is also shown. Data were analyzed with the unpaired *t*-test. Only for JUN was detected consistent changes (2.04 ± 0.22 in the tKO versus 1.05 ± 0.04 in controls; p = 0.0003) at the protein level (***p < 0.001). (**E**) Failed segregation of the axons in the Remak bundles of the tKO. A representative high power TEM image is shown. Morphometric analysis shows that axon diameter distribution is preserved in the tKO, but the number of axons per Remak bundle and the distribution of axon per pocket is changed. Five hundred axons from four animals per genotype were counted. Mixed model analysis of variance (ANOVA) with Bonferroni post hoc test was used for comparations. Scale bar: 1 μm. (**F**) Pie chart and DEG heatmap of the RNA-seq analysis of P60 showing the distribution of changed genes in the tKO. (**G**) Volcano plot shows that the most robustly changed genes were upregulated. ENSEMBL indentification numbers for the 10 most robustly changed genes are shown. (**H**) List of the 35 most upregulated genes in the adult (P60) KO classified by FDR. (**I**) List of the 35 most downregulated genes in the adult (P60) tKO classified by FDR (*p < 0.05; **p < 0.01; ***p < 0.001; ns: no significant). See source data file one online (graphs source data) for more details.

The online version of this article includes the following figure supplement(s) for figure 2:

**Figure supplement 1.** Characterizing the tKO mice.

cells was found at 10 and 20 dpi that almost normalized at 30 dpi (*Figure 4F*). Furthermore, g ratio was increased at 10 and 20 dpi but showed no difference between control and dKO nerves at 30 dpi (*Figure 4D*). Quantifying gene expression at 10 dpi demonstrated that mRNA for *Jun* remains higher in the dKO injured nerves, as does *Gdnf* (*Figure 4L*). In the same line, we found a significant decrease in the mRNA for *Krox-20*, *Periaxin*, *Mpz*, and *Mbp*. We did not find changes in the expression of *Runx2* or *Pou3f1* (*Figure 4L, M*). To substantiate these results, we looked at protein levels, and found JUN protein remained higher in the dKO at 10 dpi and was unchanged at 21 dpi. Though there was no difference in KROX-20, MPZ protein levels were decreased at 21 dpi. Together, our data support the view that the simultaneous removal of *Hdac4* and *Hdac5* from Schwann cells produces a more pronounced delay in remyelination after nerve injury.

This remyelination delay could be caused by an intrinsic problem in the capacity of Schwann cells to reactivate the myelination program, but could also be secondary to a failure in the ability of myelinating Schwann cells to acquire the repair phenotype and clear myelin debris in the distal stump. However, we did not favor the second explanation because markers of the Schwann cell repair phenotype, such as *Bdnf* and *Olig1*, in addition to *Jun*, are highly expressed in the dKO at 10 dpi (*Figure 4L, N*). Furthermore, we did not find any change in the number of intact myelin sheaths at 4 days after cut in the dKO, or in clearance of MPZ protein, suggesting no effect on the rate of demyelination (*Figure 4—figure supplement 3A–B*). Finally, repair program genes were normally upregulated in the dKO (*Figure 4—figure supplement 3C*). Together, our data show that the delay in remyelination of the dKO is due to an intrinsic defect of Schwann cells to activate the myelin transcriptional program and not a consequence of an altered reprograming capacity to the repair phenotype or to delayed myelin clearance.

Since remyelination is only moderately delayed in the dKO, we asked whether *Hdac7* was able to functionally compensate in a similar way as in development (previously, we established that cKO7 mice have no defects in myelin clearance, injury-induced gene expression or remyelination after an injury [*Figure 5—figure supplement 1*]). First, we found that tKO Schwann cells upregulate repair program genes after a cut injury (*Figure 5—figure supplement 2*) and interestingly some of them (*Olig1* and *Shh*) appear to be overexpressed at some time points (*Figure 5—figure supplement 2B, C*), suggesting that the class IIa HDACs act as a brake on the initial induction of the Schwann cell repair phenotype. In line with this observation, myelin was more rapidly cleared in these mutants (*Figure 5A–C*). Surprisingly, we could not find changes in autophagy markers neither macrophages numbers (*Figure 5—figure supplement 2G–I*), suggesting these mechanisms are not responsible of the observed increased myelin clearance.

On assessment of tKO nerves after a crush injury (*Figure 5D–N*), strikingly, we could not find any myelinated axon profile in the four tKO sciatic nerves analyzed at 10 dpi. This is in contrast to the controls which showed myelin profiles in 73 ± 3.1% of axons (p ≤ 0.0001) (*Figure 5N*). At 20 dpi, the tKO still had only 19.2 ± 5.5% of the axons myelinated, whereas almost all large caliber axons were myelinated in the control (98.1 ± 0.2%; p = 0.0001). Moreover, at 30 dpi only 60.3 ± 6.1% of the axons

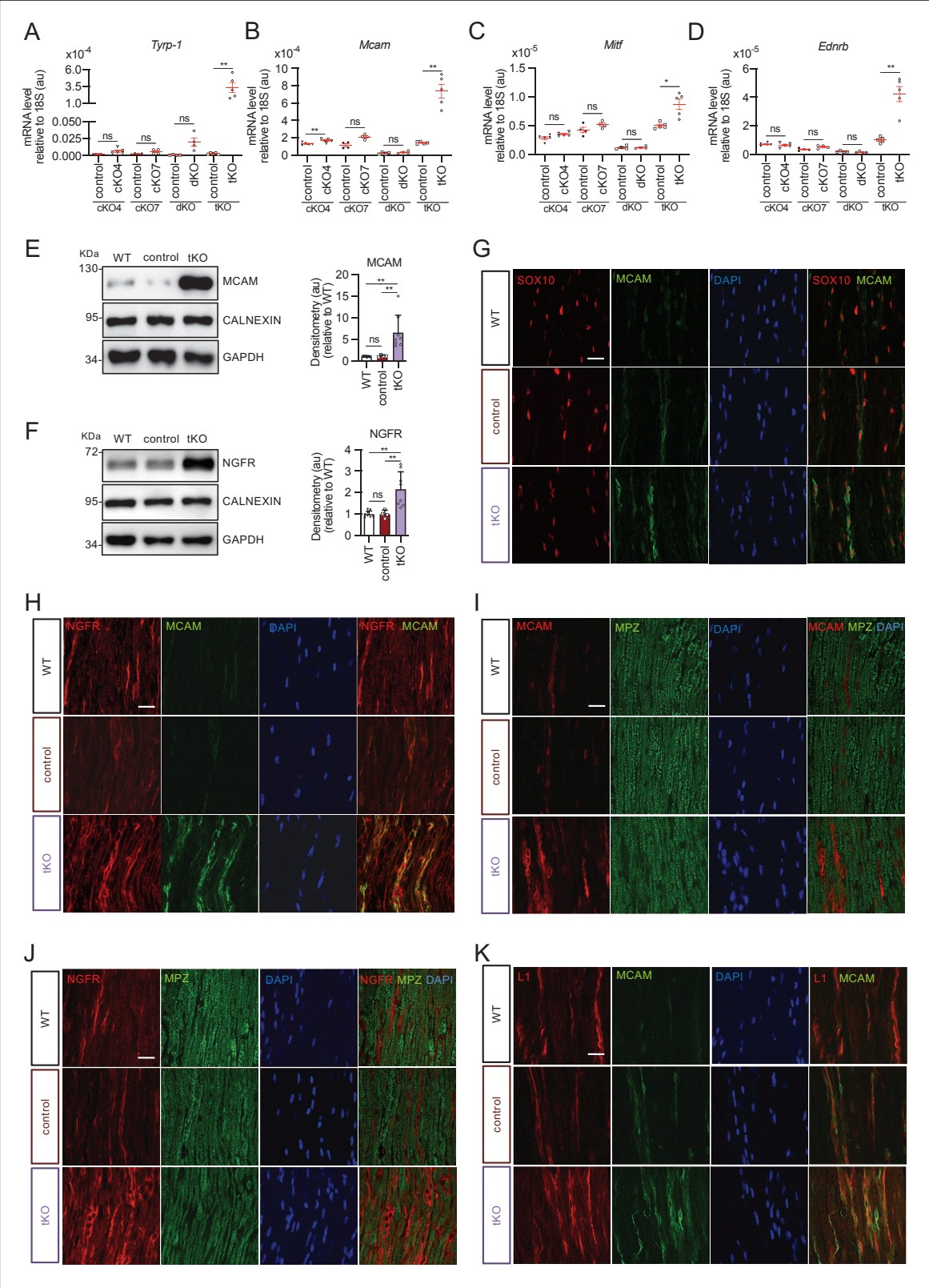

**Figure 3.** Melanocyte lineage markers are expressed by nonmyelinating Schwann cells of the Remak bundles in the sciatic nerves of the tKO. (**A**) mRNA for *Tyrp1* is dramatically increased by 1.081-fold in the tKO (3.35 ± 0.71 × 10⁻⁴ au in the tKO versus 0.11 ± 0.05 × 10⁻⁶ au in controls; p = 0.0092) whereas no changes were found in the cKO4, cKO7 neither dKO sciatic nerves. (**B**) mRNA for *Mcam* is also upregulated (5.13-fold) in the tKO (7.39 ± 0.79 × 10⁻⁴ au in the tKO versus 1.44 ± 0.06 × 10⁻⁴ au in controls) with only minor o no changes at all for the other genotypes. The same although less marked

*Figure 3 continued on next page*

Figure 3 continued

(1.74-fold) for *Mitf* ($0.87 \pm 0.09 \times 10^{-5}$ au in the tKO versus $0.50 \pm 0.02 \times 10^{-5}$ au in controls; p = 0.0128) (**C**) and *Ednrb* (4.1-fold; $4.23 \pm 0.52 \times 10^{-5}$ au in the tKO versus $1.04 \pm 0.09 \times 10^{-5}$ au in controls; p = 0.0032) (**D**). RT-qPCR with mouse-specific primers for the indicated genes was performed. Graph shows a scatter plot for the ΔCt (which include also the mean ± standard error [SE]) of the gene normalized to the housekeeping 18S. Four to five mice per genotype were used. Data were analyzed with the unpaired *t*-test with Welch's correlation. (**E**) MCAM protein levels in the sciatic nerves of the tKO. A representative Western blot of protein extracts from wild-type (C57BL/6), control and tKO sciatic nerves is shown. MCAM protein was increased by 7.6-fold in the tKO ($9.93 \pm 1.75$ au in the tKO versus $1.30 \pm 0.13$ in controls; p = 0.0003). (**F**) NGFR protein was increased by 2.15-fold ($2.16 \pm 0.29$ in the tKO versus $1.005 \pm 0.09$ in controls; p = 0.0003). Four to eight WB of the same number of animals per genotype were quantified. Data were analyzed with the one-way analysis of variance (ANOVA) Tukey's test. (**G**) MCAM signal colocalizes with SOX10. (**H**) MCAM signal colocalizes with NGFR. (**I**) MCAM is not expressed by myelin-forming Schwann cells (MPZ⁺). (**J**) Same happens with NGFR. (**K**) MCAM signal colocalizes with L1cam, a marker of the nonmyelin-forming Schwann cells of the Remak bundles. P60 sciatic nerves were fixed and submitted to immunofluorescence with the indicated antibodies. Nuclei were counterstained with Hoechst. Representative confocal images of sections obtained from the sciatic nerves of wild-type (WT), control, and tKO mice are shown. Scale bar: 20 μm (*p < 0.05; **p < 0.01; ***p < 0.001; ns: no significant). See source data file one online (graphs source data) for more details.

were myelinated (*P* = 0,0031) in the tKO mice. At 60 dpi, myelination was only slightly delayed in the tKO (*Figure 5N*). Differences in *g* ratios followed the same pattern (*Figure 5G*). In the same line, we found a notable increase in the number of unmyelinated axons >1.5 μm at 10 dpi, that decreases slowly but progressively up to 60 dpi, when it approaches a similar number to the control (*Figure 5H*). Interestingly, most of these unmyelinated axons are in a 1:1 relationship with Schwann cells (*Figure 5I*), suggesting that the delay is in the transition from the promyelinating to the myelinating Schwann cell stage. We also found a notable increase in the number of Schwann cells per nerve section that was maintained after 60 dpi (*Figure 5L*), and an increase in the nerve area (*Figure 5E*). The increased number of Schwann cells is probably consequence of over-proliferation, as suggested by Ki67 staining (*Figure 5—figure supplement 3*). We also observed a decrease in the number of axons >1.5 μm at 20 and 30 dpi (*Figure 5J*), probably because the smaller diameter of the unmyelinated axons. Finally, the delay in remyelination was substantiated by Western blot. As shown in *Figure 5O*, JUN protein is clearly more abundant in the nerves of the tKO than in either control littermates or wild-types, both at 10 and 21 dpi. Conversely, the amount of MPZ protein is lower at 10 and 21 dpi. KROX-20 is also lower at 10 dpi in the tKO, but levels had recovered by 21 dpi.

To gain insight into the functional consequences of the remyelination delay in the tKO we performed nerve impulse conduction studies of the sciatic nerves after crush injury (see Material and methods). In uninjured nerves, we found no differences in voltage amplitude or nerve conduction velocity (NCV) between tKO and controls (*Figure 6A–D*) (curiously we observed a smaller amplitude and slower NCV for all genotypes when compared with wild-type nerves (*Figure 6—figure supplement 1*), probably due to the absence of *Hdac5* in neurons). By contrast, at 40 dpi, whereas six of nine sciatic nerves of control mice showed a response when electrically stimulated at 8 V, only one of eight tKO responded (*Figure 6E, F*). The same distribution was found at 10 V. At 15 V, eight of nine control mice responded while only four of eight tKO responded. In the same line, the amplitude of the A-fiber component of the compound action potential (CAP) was decreased for 8, 10, and 15 V stimuli (*Figure 6G*). Regarding the component corresponding to C fibers, amplitude was also decreased for 8 and 10 V stimulation (*Figure 6H*). Moreover, NCV showed a statistically significant decrease when using 15 V stimuli (*Figure 6I*).

All together our data demonstrate that the kinetics of remyelination after nerve injury is directly correlated with class IIa *Hdac* gene dose.

## Targets of class IIa HDACs in Schwann cells

To try to identify the genes regulated by class IIa HDACs we first performed a genome-wide transcriptomic analysis of the tKO remyelinating sciatic nerves after a crush injury and control littermates (source data file two online [RNA-seq source data]). At 1 dpi, 395 genes were upregulated and 274 downregulated in the tKO (*Figure 5—figure supplement 4A* and *Figure 5—figure supplement 5A*). Similar to the uninjured nerve analysis, the 10 most robustly changed genes were all upregulated. Interestingly, *Tyrp1* and *Mcam* were also among the most upregulated genes. At 10 dpi, the number of dysregulated genes was notably increased, with 1227 transcripts upregulated and 1550 downregulated (*Figure 5—figure supplement 4B* and *Figure 5—figure supplement 5A*). Among the most robustly upregulated genes were the repair cell marker *Bdnf* (*Arthur-Farraj et al., 2017*; *Jessen and*

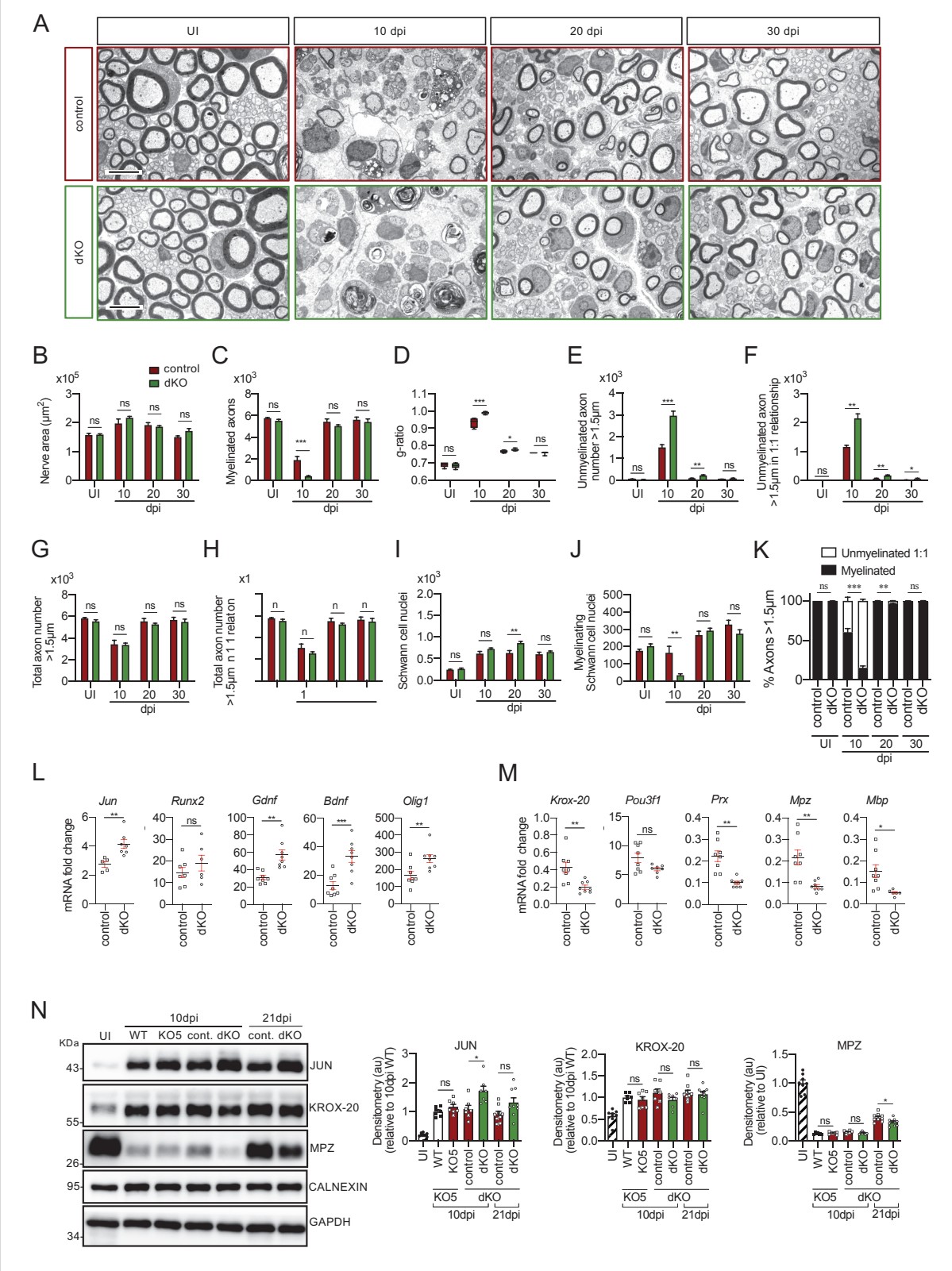

**Figure 4.** Remyelination is delayed in the nerves of the dKO mice. (**A**) Representative transmission TEM images of P60 sciatic nerves uninjured (UI) and 10, 20, and 30 days post crush (dpi) of dKO (*Mpz-Cre*[+/−]; *Hdac4*[flx/flx]; *Hdac5*[−/−]) and the control (*Mpz-Cre*[−/−]; *Hdac4*[flx/flx]; *Hdac5*[−/−]) littermates are shown. Scale bar: 5 µm. (**B**) No statistically significant differences were observed in the area of the dKO nerves and control littermates (UI: p = 0.804; 10 dpi: p = 0.195; 20 dpi: p = 0.559; 30 dpi: p = 0.0594). (**C**) The number of myelinated axons is notably decreased at 10 dpi (388 ± 55 in the dKO versus 1.889 ± 330

*Figure 4 continued on next page*

*Figure 4 continued*

in the control; p = 0.0005). (**D**) *g* ratio was increased at 10 dpi (0.989 ± 0.003 in the dKO versus 0.934 ± 0.015 in control [p = 0.002]) and at P21 (0.776 ± 0.003 in the dKO versus 0.767 ± 0.003 in control [p = 0.043]). (**E**) The number of unmyelinated axons in a 1:1 relationship with Schwann cells was notably increased at 10 dpi (2.969 ± 203 in the dKO versus 1.512 ± 119 in controls; p = 0.0007) and at 20 dpi (224 ± 25 in the dKO versus 88 ± 14 in controls; p = 0.0016). (**F**) The total number of unmyelinated axons in a 1:1 relationship with Schwann cells is increased at 10 dpi (2.148 ± 155 in the dKO versus 1.158 ± 56 in the control; p = 0.0011) at 20 dpi (175 ± 20 in the dKO versus 68 ± 12 in the control; p = 0.002) and at 30 dpi (63 ± 17 in the dKO versus 22 ± 5 in the control; p = 0.043). (**G**) No changes in the total axon number was found (UI: p = 0.157; 10 dpi: p = 0.910; 20 dpi: p = 0.349; 30 dpi: p = 0.666). (**H**) Neither in the total sorted axon number (UI: p = 0.193; 10 dpi: p = 0.169; 20 dpi: p = 0.294; 30 dpi: p = 0.682). (**I**) The total number of Schwann cells (counted as nuclei) was increased at 20 dpi (861 ± 34 in the dKO versus 630 ± 53 in controls; p = 0.0041). (**J**) In contrast, the number of myelinating Schwann cells was found decreased at 10 dpi (35 ± 8 in the dKO versus 164 ± 37 in controls; p = 0.0032). (**K**) The percentage of myelinated axons is decreased at 10 dpi (15.5 ± 2.3% in the dKO versus 60.4 ± 4.8% in controlps; p < 0.0001), 20 dpi (96.6 ± 0.4% in the dKO versus 98.8 ± 0.2% in controls; p = 0.0016) and, although much less, at P21 (98.9 ± 0.3% in the dKO versus 99.6 ± 0.1% in controls; p = 0.0482). For these experiment, three to six animals per genotype were used; unpaired *t*-test was applied for statistical analysis. (**L**) Expression of several negative regulators of myelination and repair Schwann cell markers is enhanced at 10 dpi in the sciatic nerves of the dKO: *Jun* (1.51-fold; p = 0.0056), *Gdnf* (1.85-fold; p = 0.0025), *Bdnf* (2.60-fold; p = 0.001), and *Olig1* (1.60-fold; p = 0.008). (**M**) Expression of positive regulators and myelin genes is decreased at 10 dpi in the sciatic nerves of the dKO: *Krox-20* (0.47-fold; p = 0.0068), *Prx* (0.45-fold; p = 0.001), *Mpz* (0.33-fold; p = 0.005), and *Mbp* (0.33-fold; p = 0.012). RT-qPCR with mouse-specific primers for the indicated genes was performed and normalized to 18S rRNA. The scatter plot, which include also the mean ± SE, shows the fold change of mRNA for each gene at 10 dpi normalized to the uninjured nerve. Five to eight mice per genotype were used. Data were analyzed with the unpaired *t*-test with Welch's correlation. (**N**) A representative WB of protein extracts from dKO, control, KO5$^{-/-}$ and wild-type nerves is shown. In the quantification, JUN protein remains higher in the dKO at 10 dpi (1.72 ± 0.17-fold; p = 0.012) and tend to equalize at 21 dpi. MPZ protein was found decreased by 0.32 ± 0.02-fold at 21 dpi (p = 0.02), however we could not find changes in *KROX-20* (KO5$^{-/-}$ mice were used to compare with the wild-type littermates). Densitometric analysis was done on seven to nine WB from the same number of mice and normalized to 10 dpi WT. Data were analyzed with the unpaired *t*-test (*p < 0.05; **p < 0.01; ***p < 0.001; ns: no significant). See source data file one online (graphs source data) for more details.

The online version of this article includes the following figure supplement(s) for figure 4:

**Figure supplement 1.** Remyelination in the cKO4 mice.

**Figure supplement 2.** Remyelination in the KO5 mice.

**Figure supplement 3.** Myelin clearance and repair phenotype activation in the dKO mice.

*Mirsky, 2019*), possibly reflecting impairment of the tKO repair Schwann cells to fully redifferentiate into myelin and nonmyelin-forming Schwann cells. Interestingly, in contrast with previous time points, 8 of the 10 most robustly changed genes were downregulated, including many myelin-related genes (*Mal, Prx, Mag, Ncmap, Pmp22, Mbp*, and *Mpz*), which is expected given the dramatic delay in myelin sheath development in the tKO at 10 dpi (*Figure 5N*). At 20 dpi, we identified 1895 transcripts upregulated and 2450 downregulated in the tKO (*Figure 5—figure supplement 4C* and *Figure 5—figure supplement 5A*). As before, the 10 most robustly changed genes were upregulated and among them we found again *Tyrp1, Mcam*, and *Ednrb* (*Figure 5—figure supplement 4C*). The negative regulator of myelination *Jun* was upregulated in the tKO from 1 dpi and remained increased up to 20 dpi (*Figure 5—figure supplement 5B*). A similar profile was shown for *Runx2, Gdnf, Ngfr*, and *Sox2* (*Figure 5—figure supplement 5C–F*). *Pou3f1* was induced up to 10 dpi in both control and tKO nerves, to later (20 dpi) be downregulated in the control (20 dpi) but not in the tKO nerves (*Figure 5—figure supplement 5G*). By contrast, the master transcriptional regulator of myelination *Krox-20* was downregulated at 10 and 20 dpi in the tKO, when remyelination was highly active in the controls (*Figure 5—figure supplement 5H*). Probably as a consequence, early myelin genes such as *Drp2* and *Prx* were downregulated (*Figure 5—figure supplement 5I, J*). Other myelin protein genes that are expressed later, such as *Mpz, Mbp, Mag, Pmp22*, and *Plp1* were also consistently downregulated (*Figure 5—figure supplement 5K–O*). Myelin is a specialized plasma membrane with a distinctive lipid composition particularly rich in cholesterol (*Poitelon et al., 2020*). During remyelination of the tKO nerves we found several genes of the sterol branch of the mevalonate pathway downregulated (*Hmgcs, Lss*, and *Dhcr24*) (*Figure 5—figure supplement 5P–R*). We also found downregulated genes encoding for enzymes involved in the elongation (*Elovl1*), transport (*Pmp2*) and insertion of double bonds (*Scd2* and *Fads1*) into fatty acids (*Figure 5—figure supplement 5S–V*). Interestingly, *Cers2* and *Ugt8a* were also downregulated. These genes are involved in the synthesis of sphingomyelin and galactosyl-ceramide, respectively (*Figure 5—figure supplement 5W, X*), both abundant lipids in myelin (*Poitelon et al., 2020*). Together our data show that class IIa HDACs are necessary to block negative regulators of myelination and induce the expression of genes encoding for myelin proteins and key enzymes for the biosynthesis of myelin lipids.

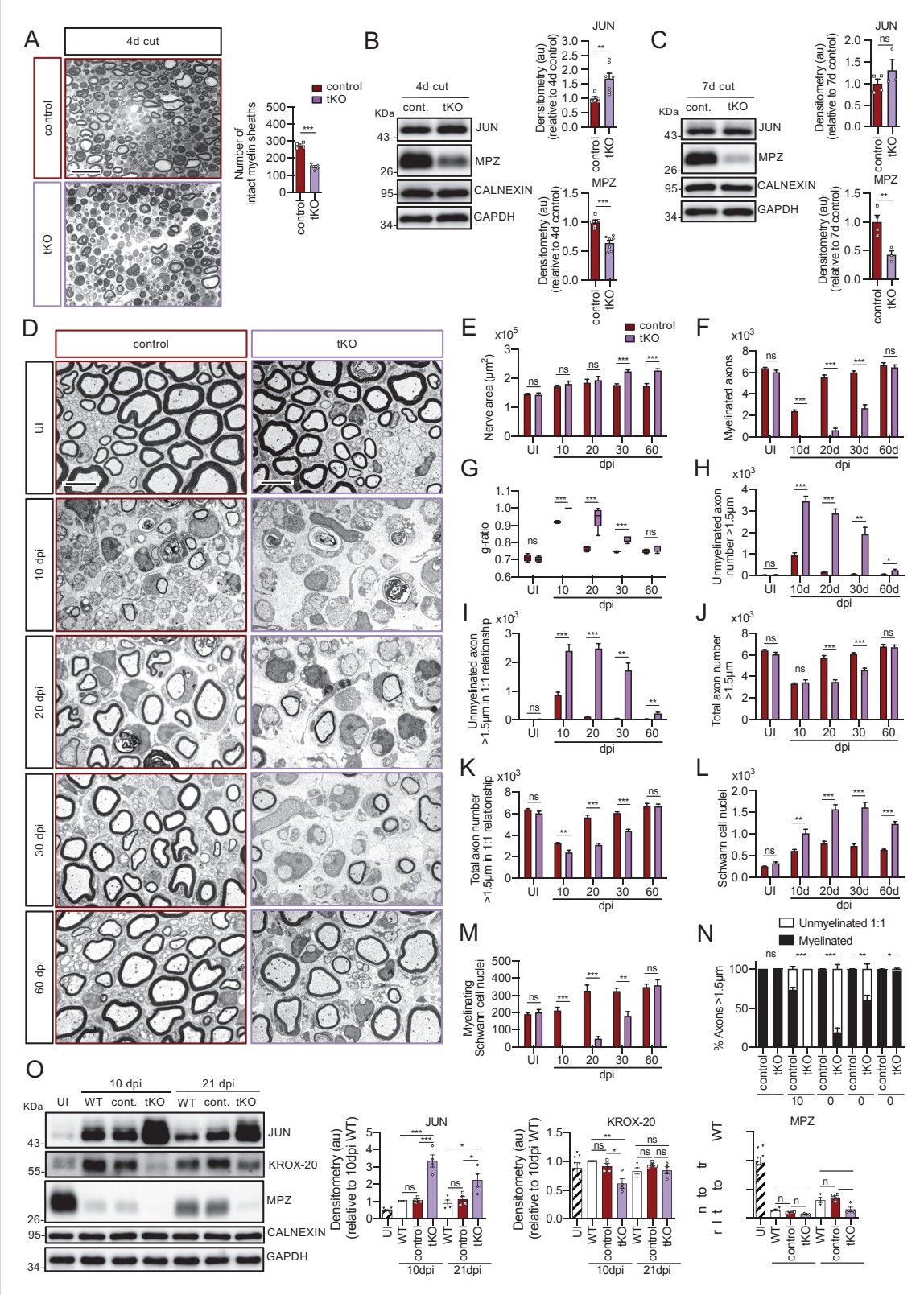

**Figure 5.** Remyelination is dramatically delayed in the tKO. (**A**) Myelin clearance is accelerated in the sciatic nerves of the tKO. A representative toluidine blue staining image of 4 days cut sciatic nerve of tKO and control mice is shown. The quantification of intact myelin sheaths shows a 0.55-fold change (p < 0.0001) in the tKO (150 ± 7 intact myelin sheaths in the dKO versus 274 ± 8 in controls; p < 0.0001). Seven to eight animals per genotype were used for the experiment. Data were analyzed with the unpaired *t*-test. Scale bar: 10 μm. (**B**) WB supports an accelerated myelin clearance in the

*Figure 5 continued on next page*

Figure 5 continued

tKO. As is shown, MPZ protein is decreased (0.7-fold; p = 0.0062) in the tKO. As expected, JUN was increased. Four mice per genotype were used. (**C**) At 7 days post-cut the decrease in the MPZ protein was more marked (0.43-fold; p = 0.012). Three to four mice per genotype were used. CALNEXIN and GAPDH were used as housekeepings. Densitometric analysis was preformed and normalized to the controls. Data were analyzed with the unpaired *t*-test. (**D**) Representative transmission TEM images of P60 sciatic nerves uninjured (UI) and 10, 20, 30, and 60 days post crush (dpi) of tKO and the control littermates are shown. Scale bar: 5 µm. (**E**) No statistically significant differences were observed for the area of the tKO nerves and control littermates in the UI (p = 0.897), at 10 dpi (p = 0.456) neither at 20 dpi (p = 0.602). The nerve area of the tKO was increased at 30 dpi (1.26-fold; p = 0.0002) and at 60 dpi (1.30-fold; p = 0.0008). (**F**) No myelinated axons in the tKO were found at 10 dpi whereas the control had 2.383 ± 112 axons myelin (p < 0.0001). Myelinated axons were decreased at 20 dpi (606 ± 200 in the tKO versus 5.525 ± 222 in the control; p < 0.0001), at 30 dpi (2.659 ± 323 in the tKO versus 6.003 ± 125 in the control; p < 0.0001), to catch up at 60 dpi (6.458 ± 240 in the tKO versus 6.689 ± 212 in the control: p = 0.491). (**G**) *g* ratio was increased in the tKO at 10 dpi (1 ± 0 in the tKO versus 0.92 ± 0.003 in control; p < 0.0001) at 20 dpi (0.94 ± 0.027 in the tKO versus 0.77 ± 0.006 in control; p = 0.0023), and at 30 dpi (0.82 ± 0.008 in the tKO versus 0.75 ± 0.002 in control; p = 0.0009). (**H**) The number of unmyelinated axons >1.5 was increased at 10 dpi (3.477 ± 236 in the tKO versus 950 ± 116 in controls; p < 0.0001), at 20 dpi (2.885 ± 209 in the tKO versus 184 ± 15 in controls; p = 0.0002), at 30 dpi (1.925 ± 319 in the tKO versus 76 ± 16 in controls; p = 0.0044), and at 60 dpi (257 ± 43 in the tKO versus 69 ± 11 in controls; p = 0.010). (**I**) The number of unmyelinated axons >1.5 in a 1:1 relationship with Schwann was notably increased at 10 dpi (2.405 ± 209 in the tKO versus 864 ± 102 in controls; p = 0.0006), at 20 dpi (2.487 ± 170 in the tKO versus 110 ± 13 in controls; p = 0.0001), at 30 dpi (1.728 ± 250 in the tKO versus 43 ± 8 in controls; p = 0.0025), and at 60 dpi (224 ± 43 in the tKO versus 28 ± 5 in controls; p = 0.010). (**J**) In contrast, the total number of axons >1.5 µm was decreased at 20 dpi (3.492 ± 184 in the tKO versus 5.790 ± 228 in the control; p < 0.0001) and at 30 dpi (4.584 ± 184 in the tKO versus 6.080 ± 131 in the control; p = 0.0002) to finally catch up at 60 dpi (6.716 ± 198 in the tKO versus 6.758 ± 221 in the control; p = 0.889). (**K**) The total number of axons >1.5 µm sorted (in a 1:1 relationship with Schwann cells) was decreased at 10 dpi (2.405 ± 209 in the tKO versus 3.247 ± 60 in the control; p < 0.0082), at 20 dpi (3.093 ± 147 in the tKO versus 5.653 ± 233 in the control; p < 0.0001), and at 30 dpi (4.387 ± 158 in the tKO versus 6.046 ± 127 in the control; p < 0.0001). (**L**) The number of Schwann cells (counted as nuclei) was increased at 10 dpi (1.017 ± 95 in the tKO versus 609 ± 33 in the control; p < 0.0001), at 20 dpi (1.576 ± 100 in the tKO versus 779 ± 54 in the control; p < 0.0001), and at 30 dpi (1.618 ± 116 in the tKO versus 723 ± 47 in the control; p < 0.0001). (**M**) The number of myelinating Schwann cells was also decreased at 10 dpi (0 ± 0 in the tKO versus 212 ± 18 in the control; p < 0.0001), at 20 dpi (48 ± 12 in the tKO versus 326 ± 33 in the control; p < 0.0001), and at 30 dpi (181 ± 24 in the tKO versus 325 ± 17 in the control; p = 0.0011). (**N**) The percentage of myelinated axons is decreased at 10 dpi (0 ± 0% in the tKO versus 73.1 ± 3.1% in controls; p < 0.0001), 20 dpi (19.2 ± 5.5% in the tKO versus 98.1 ± 0.2% in controls; p < 0.0001), 30 dpi (60.3 ± 6.1% in the tKO versus 99.3 ± 0.1% in controls; p = 0.0031), and at 60 dpi (96.6 ± 0.7% in the tKO versus 99.6 ± 0.1% in controls; p < 0.0001). For these experiments, four to five animals per genotype were used; unpaired *t*-test was applied for statistical analysis. (**O**) A representative WB of protein extracts from tKO, control and wild-type nerves is shown. In the quantification, JUN protein remains higher in the tKO at 10 dpi (3.24 ± 0.35-fold; p < 0.0001) and at 21 dpi (2.25 ± 0.38-fold; p = 0.031). *KROX-20* was found decreased at 10 dpi (0.61 ± 0.08-fold; p = 0.011). MPZ protein was found decreased at 21 dpi by 0.15 ± 0.04-fold dpi (p = 0.0091). Densitometric analysis was done for seven to nine WB from the same number of mice and normalized to 10 dpi WT. Data were analyzed with the unpaired *t*-test (*p < 0.05; **p < 0.01; ***p < 0.001; ns: no significant). See source data file one online (graphs source data) for more details.

The online version of this article includes the following figure supplement(s) for figure 5:

**Figure supplement 1.** Remyelination in the cKO7 mice.

**Figure supplement 2.** Repair Schwann cell phenotype and myelin removal in the injured tKO.

**Figure supplement 3.** Increased Jun and Schwann cell proliferation in the tKO sciatic nerve.

**Figure supplement 4.** Evolution of sciatic nerve gene expression profile at 1, 10, and 20 days post crush (dpi).

**Figure supplement 5.** Time course analysis of myelination-relevant gene expression during nerve regeneration from RNA-seq.

To learn which genes are direct targets of class IIa HDACs and which are regulated indirectly, we performed a chromatin immunoprecipitation assay with anti-HDAC4 coupled to massive sequencing (ChIP-Seq) in dbcAMP differentiated Schwann cells. We found 3.932 peaks, 67.27% of which were located in the proximal promoter regions of genes (≤1 kb from the transcription start site [TSS]) (*Figure 7A*). The localization of these peaks in the rat genome is shown in the source data file three online (ChIP-Seq peaks source data). Importantly, ChIP-Seq analysis confirmed our previous results (*Gomis-Coloma et al., 2018*) showing that HDAC4 binds to the promoters of *Jun*, *Gdnf*, and *Runx2* (*Figure 7B–E*). Interestingly, HDAC4 also binds to the promoter region of *Sox2* (*Figure 7B, F*), another negative regulator of myelination. We found also peaks for *Id2* and *Hey2* (*Figure 7B*) and source data file three online (ChIP-Seq peaks source data).

Here, we show that *Pou3f1* is overexpressed in the PNS of the tKO during development (*Figure 1K–M*), and that it is not properly downregulated during remyelination (*Figure 5—figure supplement 5G*). Interestingly, we found three peaks of HDAC4 bound near the TSS of *Pou3f1* (*Figure 7B–G*), a result that was confirmed by ChIP-qPCR (*Figure 7H*).

Regarding the melanocyte lineage, we found a clear peak of HDAC4 close the TSS of *Mcam* (*Figure 7B*), however, we did not detect peaks in *Tyrp1* and *Ednrb*, suggesting that, while HDAC4

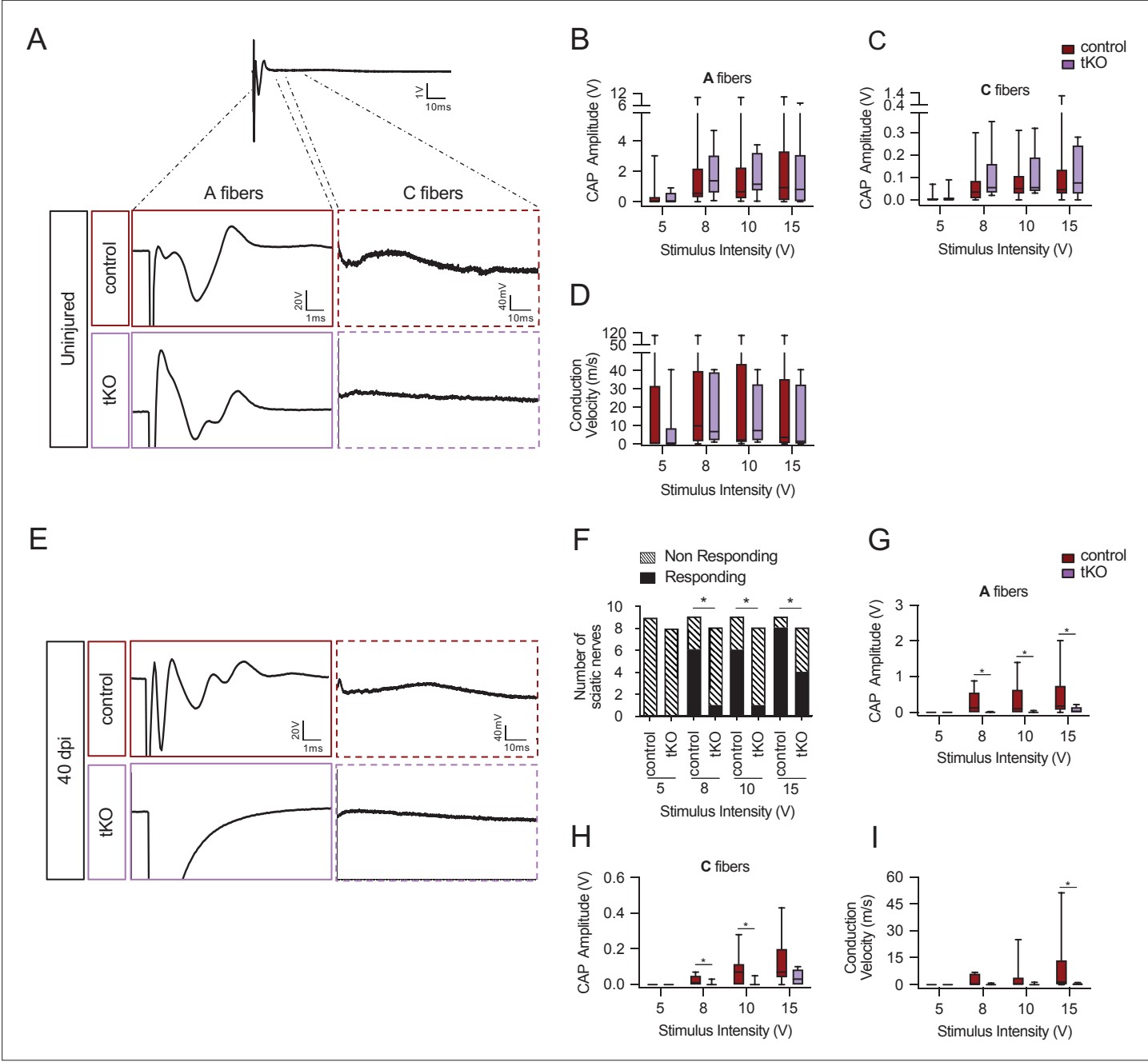

**Figure 6.** Remyelination failure in the tKO hampered nerve impulse conduction. (**A**) Sample recordings of compound action potential (CAP) in uninjured sciatic nerves of control and tKO mice showing the waveform components corresponding to myelinated (A-fibers) and unmyelinated (**C**) fibers. (**B, C**) Waveform component of A-fibers (**B**) and C-fibers (**C**) showed similar amplitude in both genotypes for stimulation with pulses of increasing intensity (5, 8, 10, and 15 V). (**D**) Nerve conduction velocity was also preserved in the tKO mice. (**E**) Sample recordings of CAPs obtained in a control and a tKO sciatic nerve 40 days after nerve crushing. (**F**) The number of nerves that responded with a detectable CAP after the stimulation with increasing intensity. (**G**) Amplitude of CAP corresponding to A-fibers was significantly smaller in tKO than in control nerves for stimulation with 8, 10, and 15 V. (**H**) Amplitude of C-fiber component was significantly smaller in tKO than in control nerves for stimulation with 8 and 10 V. (**I**) Nerve conduction velocity was significantly reduced after injury in both genotypes, being significantly lower in tKO than in control nerves for stimulation at 15 V. In this set of experiments, the whole length of a sciatic nerve was exposed from its proximal projection (L4 spinal cord) to its distal branches in deeply anesthetized mice. Compound action potentials (CAPs) were evoked by electrical stimulation of increasing intensity (5, 8, 10, and 15 V, 0.03-ms pulse duration). The maximum amplitude of the A- and C-fiber components of CAP electrical signal, and their mean nerve conduction velocity were measured. Seven to 18 animals per genotype and condition were used. Mann–Whitney's $U$ was used for nonparametric paired comparisons and chi-squared test was used for statistical comparations (*$p < 0.05$; **$p < 0.01$; ***$p < 0.001$; ns: no significant). See source data file one online (graphs source data) for more details.

*Figure 6 continued on next page*

*Figure 6 continued*

The online version of this article includes the following figure supplement(s) for figure 6:

**Figure supplement 1.** HDAC5 elimination decreases nerve conduction velocity and voltage amplitude.

directly represses the expression of *Mcam* it likely indirectly leads to *Tyrp1* and *Ednrb* repression. However, an alternative explanation is that it achieves repression of these genes by using alternative promoters or enhancers.

Surprisingly, we also found peaks in the promoter regions of *Mbp* and *Hmgcr* (**Figure 7B**), two genes highly expressed during myelination. Although it could seem contradictory as HDACs have mainly been described as transcriptional repressors, HDACs have been shown to be bound to the promoter regions of highly expressed genes in other tissues (**Wang et al., 2009**).

## JUN binds to the promoter and induces the expression of *Hdac7* gene in the PNS

As we have shown, the simultaneous elimination of *Hdac4* and *Hdac5* activates a mechanism to compensate for the drop in class IIa *Hdac* gene dose in Schwann cells. This mechanism multiplies by threefold the expression of *Hdac7*, the other member of this family expressed in these cells (**Figure 1A**). But what mechanism is involved? We have shown before that *Jun* is highly expressed in the developing PNS of the dKO mice (**Gomis-Coloma et al., 2018**). Here, we show that *Jun* expression remains high during remyelination in the dKO sciatic nerves (**Figure 4L, N** and **Figure 5—figure supplement 5B**). Interestingly, we found in ENCODE (https://www.encodeproject.org/) that JUN binds to the promoter region of *Hdac7* in A549 cells. Thus, the increased JUN might bind to the promoter of *Hdac7* inducing its compensatory overexpression in the dKO nerves. To test this, we used ChIP-qPCR and found that, indeed, in cultured Schwann cells JUN binds to the *Hdac7* promoter (**Figure 8A**).

We then identified an evolutionarily conserved JUN consensus binding motif in the proximal promoter region of the mouse *Hdac7* gene (**Figure 8—figure supplement 1**). To test it functionally, a fragment of 1.189 bp containing this region was amplified by PCR and cloned into the pGL3-luciferase reporter vector (see Materials and methods). The 1.189-promoter-*Hdac7*-pGL3 luciferase construct was transfected into HEK293 cells together with a pcDNA3 plasmid encoding for *Jun*, and luciferase activity measured 12 hr post-transfection. As shown in **Figure 8B**, this promoter fragment responded to JUN by increasing the luciferase activity by 3.4-fold over the control, which supports the idea that *Hdac7* expression is regulated by JUN.

To further test this hypothesis in vivo, we utilized mouse transgenic lines (*Mpz-Cre*[+/−]/*Rosa26*[flox-stop-Jun/+] mice) that either overexpresses *Jun* in Schwann cells, referred to as Jun_OE mice (**Fazal et al., 2017**) or lack *Jun* expression in Schwann cells (*Mpz-Cre*[+/−]/*Jun*[flox/flox] mice, referred to as Jun_cKO mice; **Parkinson et al., 2008**; **Figure 8C**). *Jun* overexpression in Schwann cells induced *Hdac7* expression by almost twofold (**Figure 8D**). This was a specific effect as no changes were found for *Hdac4* and *Hdac5* mRNA (**Figure 8E, F**). By contrast, *Jun* removal produced no changes in the expression of any class IIa *Hdacs*, suggesting it is not necessary for the basal expression of these *Hdacs* (**Figure 8D–F**). Thus, our data clearly show that JUN induces the expression of *Hdac7* by Schwann cells in vivo. Further supporting this tenet, we found that *Hdac7* is not overexpressed in the nerves of the dKO that lack *Jun* in Schwann cells (dKO;Jun_cKO) (**Figure 8G**).

Interestingly, we detected a peak of HDAC4 bound to the promoter of *Hdac7* (**Figures 7B and 8H**) in the ChIP-Seq experiment, a result that we confirmed by ChIP-qPCR (**Figure 8I**). This suggests that, in differentiated Schwann cells, other class IIa HDACs contribute to maintaining *Hdac7* expression at the basal level. The simultaneous loss of *Hdac4* and *Hdac5* allows the expression of *Jun*, which can bind to the promoter of *Hdac7*, now free of the repression by class IIa HDACs, increasing the transcription of this deacetylase.

## *Hdac9* is expressed de novo in the sciatic nerve of the tKO

Despite substantial delay, Schwann cells in tKO nerves are still able to eventually myelinate axons in development and during nerve repair. One possible explanation is that *Hdac9*, the remaining class IIa *Hdac* left, could compensate for the loss of the other three class IIa *Hdacs* in Schwann cells. However, we have consistently found extremely low or undetectable levels of the mRNA for this protein in

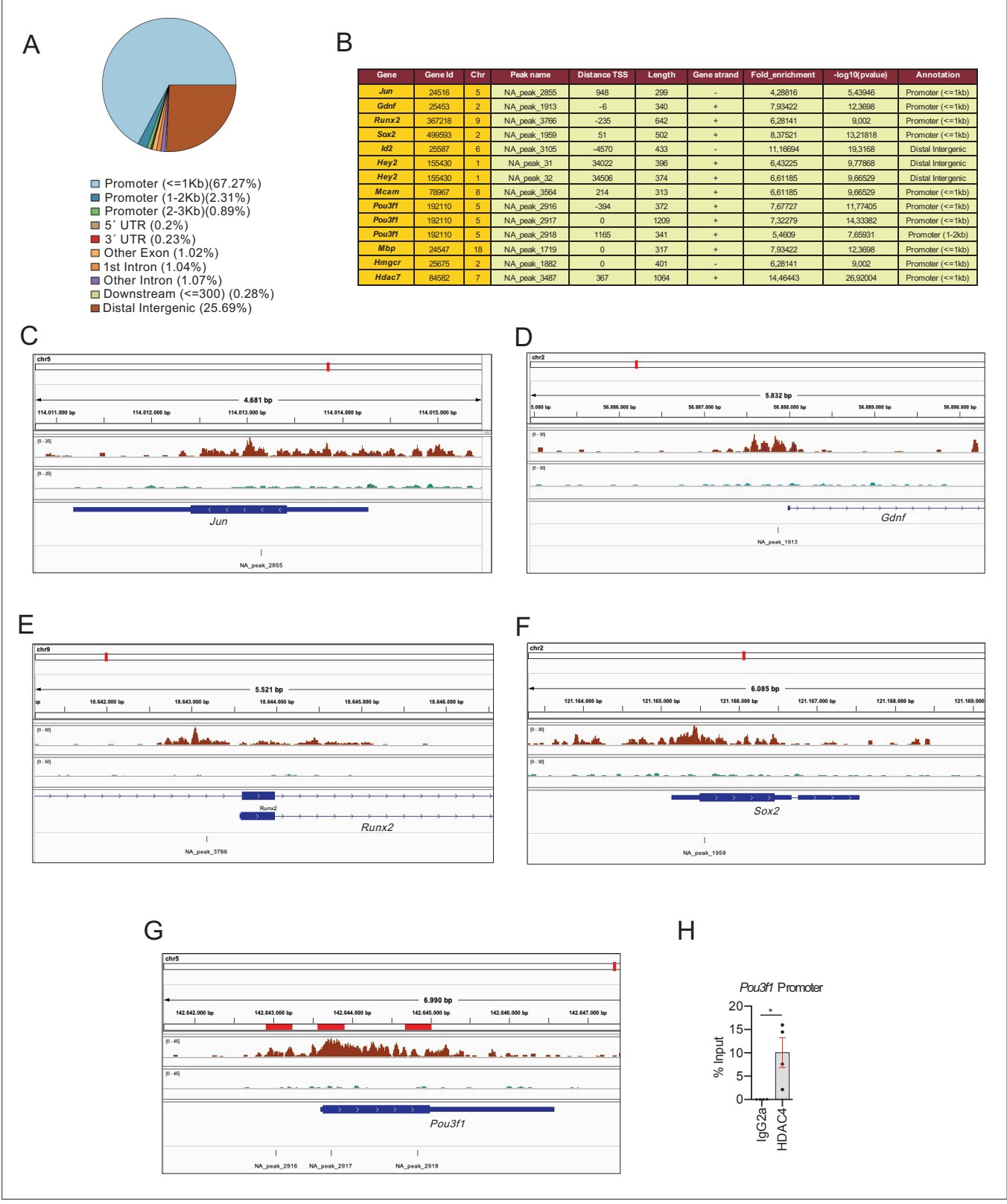

**Figure 7.** HDAC4 binds to the promoter of pivotal genes for myelin development. Cultured rat Schwann cells were incubated with 1 mM dbcAMP to shuttle HDAC4 into the nucleus, and ChIP-Seq analysis performed on the crosslinked chromatin, with anti-HADC4 antibody. (**A**) 3.932 HDAC4 peaks were found in the rat genome. Most of these peaks (67.27%) were located in the promoter regions (≤1 kb from the TSS). (**B**) A table with the localization of some of these peaks (a complete list can be found in source data file three online [ChIP-Seq peaks source data]). (**C–E**) ChIP-Seq signal analysis

*Figure 7 continued on next page*

Figure 7 continued

confirmed our previous results (**Gomis-Coloma et al., 2018**) showing that HDAC4 binds to the promoters of *Jun*, *Gdnf*, and *Runx2*. (**F**) HDAC4 was also found bound to the promoter region of *Sox2*. (**G**) Three peaks (NA_peaks 2916, 2917, and 2918) near the TSS of *Pou3f1* were also found. (**H**) The binding of HDAC4 to *Pou3f1* gene was confirmed by ChIP-qPCR. Four different experiments from four distinct cultures were used. Data were analyzed with the Mann–Whitney test (*p < 0.05; **p < 0.01; ***p < 0.001; ns: no significant). See source data file one online (graphs source data) for more details.

the C57BL/6 mouse sciatic nerves and cultured rat Schwann cells (**Gomis-Coloma et al., 2018**), a result that has been recently confirmed in the Sciatic Nerve ATlas (SNAT, https://www.snat.ethz.ch) (**Gerber et al., 2021**). Surprisingly, we found *Hdac9* among the most robustly upregulated genes by the tKO nerves in the RNA-seq analysis (**Figure 2H**). To confirm this result, we measured the mRNA for this gene in the sciatic nerves of the tKO and control adult mice (P60) by RT-qPCR. As is shown in **Figure 9A**, the mRNA for *Hdac9* was increased by 4.4-fold in the tKO mice (p < 0.0001). This increase was not found in the dKO and was smaller in the cKO7 (2.4-fold; p = 0.0057) and cKO4 (1.3-fold; p = 0.01) mice. These results suggest that *Hdac9* is strongly expressed in the sciatic nerves of the tKO mice to compensate for the absence of other class IIa HDACs. Indeed, we found that *Hdac9* is already induced early in development in the tKO (P2) whereas it was practically undetectable in the nerves of control animals (**Figure 9B**). At this time, myelin sheaths are present, although sparse, in mutant nerves (**Figure 1B–J**). At P8, *Hdac9* expression remains extremely low in control nerves, but it is robustly expressed in the tKO mice (an increase of 5.3-fold; p = 0.0117), correlating with an increase in the number of myelinated axons (**Figure 1B–J**). Robust *Hdac9* gene expression is maintained in the sciatic nerve of the tKO at P60 (**Figure 9A**). Interestingly, we found that in the tKO nerves, the lysine 9 of histone three associated to the *Hdac9* promoter is more acetylated (**Figure 9C**), suggesting that the absence of class IIa HDACs allows the expression of this gene. Importantly, *Hdac9* gene is furtherly upregulated in the injured sciatic nerves of tKO mice, as is shown in **Figure 9D**, where we show the alignment of the reads of the RNA-seq from three individual sciatic nerves from control and tKO mice UI and at 20 dpi, and in **Figure 9E**, where we followed the mRNA levels of *Hdac9* (as Fragments Per Kilobase Million, FPKMs) at 0, 1, 10, and 20 days post crush in the RNA-seq experiment. Thus far, our data suggested that in the absence of class IIa *Hdacs*, repression is lost and Schwann cells start to express de novo Hdac9.

To investigate which transcription factor is responsible of *Hdac9* expression in the tKO we again looked at JUN. However, we could not detect JUN bound to the promoter region of *Hdac9* in A549 cells in the ENCODE database. Also, *Hdac9* mRNA levels were not increased in the sciatic nerves of the Jun_OE mice (**Figure 9—figure supplement 1**).

Interestingly, it has been described that MEF2D binds to the promoter and regulates the expression of *Hdac9* during muscle differentiation, and in leiomyosarcoma cells (**Di Giorgio et al., 2020**; **Haberland et al., 2007**). We therefore reasoned that *Mef2d* might be involved in the upregulation of *Hdac9* in Schwann cells. Indeed, we have previously shown that *Mef2* family members are expressed in cultured Schwann cells and adult sciatic nerves (**Gomis-Coloma et al., 2018**). Importantly, RT-qPCR shows that *Mef2d* mRNA is induced in the nerves of the tKO during development (**Figure 9F**). Also, RNA-seq data showed that *Mef2d* is upregulated in the sciatic nerves of the tKO mice at 10 and 20 dpi (**Figure 9G**). Western blot supported this idea (**Figure 9H, I**) and immunofluorescence studies (**Figure 9J**) showed that *Mef2d* is expressed by Schwann cells (SOX10[+]). Importantly, we detected MEF2D bound to the promoter of *Hdac9* in the tKO nerves at 20 dpi (**Figure 9K**).

Altogether our data support the view that in the tKO nerve, *Mef2d* is induced to activate the de novo expression of *Hdac9* and maintain a sufficient class IIa *Hdac* gene dose to allow myelin formation during development and after nerve injury.

## Discussion

Functional redundancies, the consequence of gene duplications, are found in most genomes, and are postulated to give robustness to organisms against mutations. However, it has been also predicted that redundancies are evolutionarily unstable and have only a transient lifetime. Despite of this, numerous examples exist of gene redundancies that have been conserved throughout dilated evolutionary periods (**Kafri et al., 2006**; **Peng, 2019**).

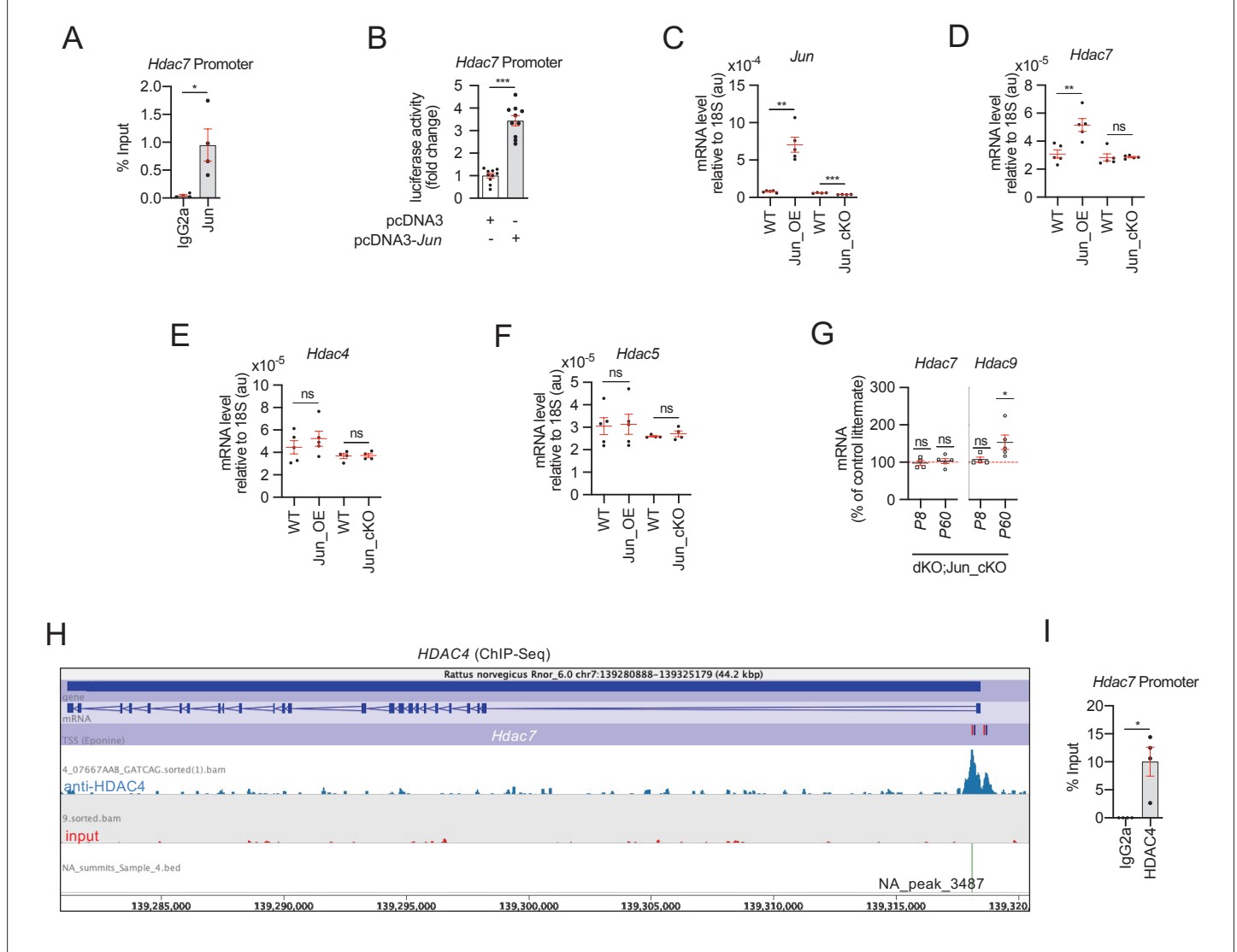

**Figure 8.** HDAC7 compensatory overexpression is induced by JUN. (**A**) ChIP-qPCR of dbcAMP treated rat Schwann cells with anti-JUN antibody showed that this transcription factor is bound to the promoter of *Hdac7*. Four different experiments from four distinct cultures were used. Data were analyzed with the Mann–Whitney test. (**B**) A 1.189 fragment of the mouse *Hdac7* promoter containing a conserved JUN binding consensus sequence was PCR amplified and cloned into the pGL3 luciferase reporter vector. HEK293 cells were transfected with this construct and the pcDNA3 (empty vector) or pcDNA3 *Jun*. As is shown JUN induced the luciferase activity by 3.4 ± 0.22-fold (p < 0.0001; *n* = 10). Unpaired *t*-test with Welch's correlation was used for statistical comparison. (**C**) Levels of mRNA for *Jun* in the nerves of Jun_OE and Jun_cKO mice. (**D**) *Hdac7* expression is enhanced in the sciatic nerves of the Jun_OE mice (5.16 ± 0.46 × 10⁻⁵ in the Jun_OE versus 3.09 ± 0.29 × 10⁻⁵ in the WT; p = 0.005) but does not change in the Jun_cKO mice. (**E**) *Hdac4* expression in sciatic nerves does not change in Jun_OE and Jun_cKO mice. (**F**) *Hdac5* expression in sciatic nerves does not change in Jun_OE and Jun_cKO mice. (**G**) Removal of *Jun* from Schwann cells in the dKO (dKO;Jun_cKO genotype) prevents *Hdac7* compensatory overexpression. Interestingly, *Hdac9* expression is induced in these mice (see discussion). RT-qPCR with mouse-specific primers for the indicated genes was performed. The scatter plot, which include also the mean ± standard error (SE), shows the expression of each gene normalized to the housekeeping 18S. Four to five mice per genotype were used. Data were analyzed with the unpaired *t*-test with Welch's correlation. (**H**) A peak of HDAC4 (NA_peak 3487) was found on the *Hdac7* promoter in the ChIP-Seq experiemt. (**I**) ChIP-qPCR confirmed that HDAC4 is bound to the promoter of *Hdac7*. Four different experiments from four distinct cultures were used. Data were analyzed with the Mann–Whitney test (*p < 0.05; **p < 0.01; ***p < 0.001; ns: no significant). See source data file one online (graphs source data) for more details.

The online version of this article includes the following figure supplement(s) for figure 8:

**Figure supplement 1.** The proximal promoter region of the *Hdac7* has a JUN consensus binding sequence.

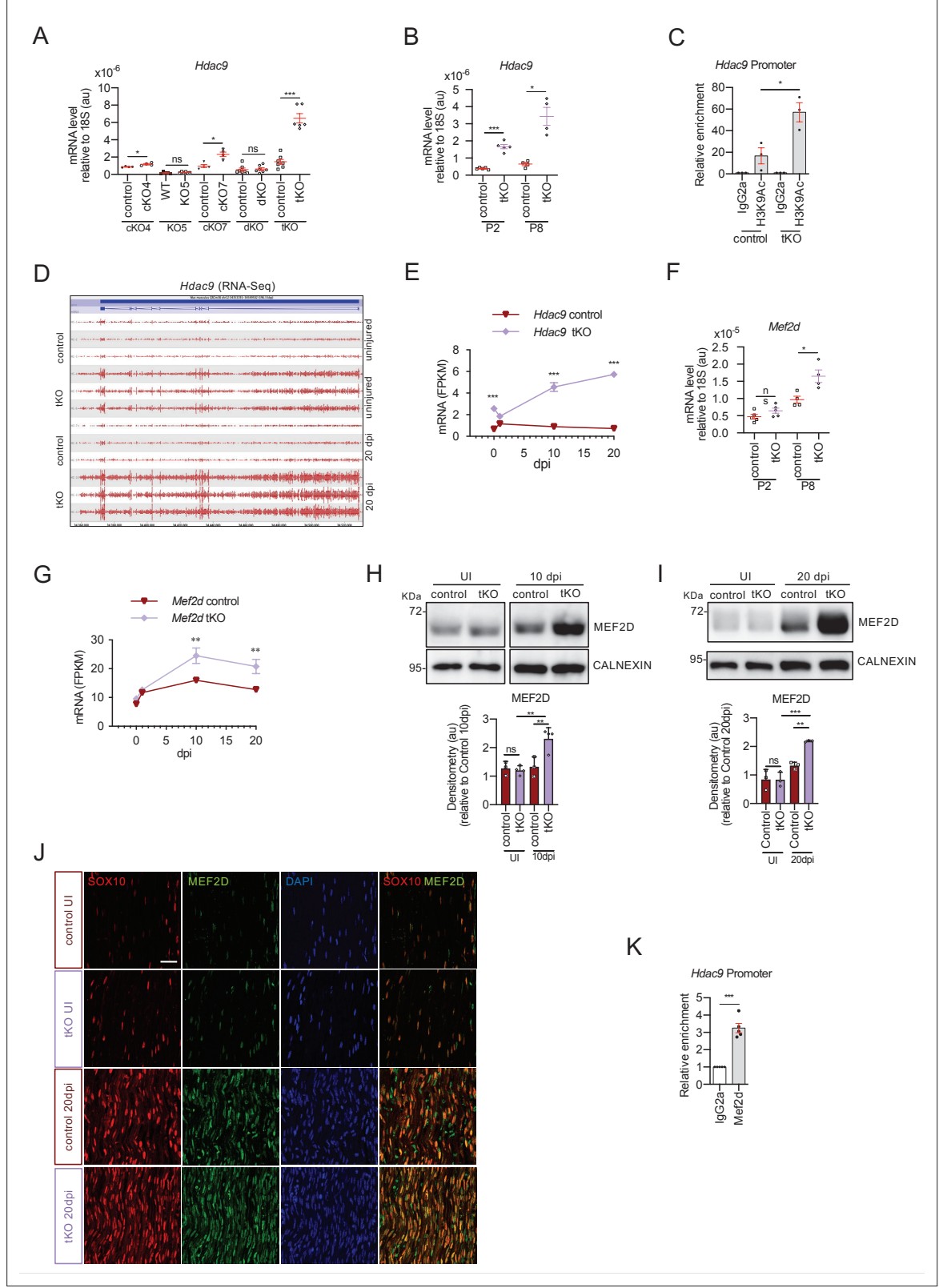

**Figure 9.** MEF2D mediates *Hdac9* de novo expression in the tKO. (**A**) *Hdac9* expression is notably induced in the sciatic nerves of the adult (P60) tKO mice (6.48 ± 0.53 × 10$^{-6}$ au the tKO versus 1.46 ± 0.28 × 10$^{-6}$ in the control; p < 0.0001). Only minor changes were observed in the cKO4 and cKO7. Four to eight mice per genotype were used. Unpaired *t*-test was used for comparations. (**B**) *Hdac9* expression is increased from early postnatal development of the tKO nerve. At P2 we found 1.67 ± 0.13 × 10$^{-6}$ au in the tKO versus 0.39 ± 0.03 × 10$^{-6}$ in the controls (p < 0.0001) and at P8 we found 3.43 ± 0.52 ×

*Figure 9 continued on next page*

*Figure 9 continued*

$10^{-6}$ au in the tKO versus $0.65 \pm 0.09 \times 10^{-6}$ in the controls (p < 0.0001). RT-qPCR with mouse-specific primers for *Hdac9* was performed. The scatter plot, which include also the mean ± standard error (SE), shows the expression of *Hdac9* normalized to the housekeeping 18S. Four to five mice per genotype were used. Data were analyzed with the unpaired *t*-test with Welch's correlation. (**C**) ChIP-qPCR with anti-H3K9Ac of adult (P60) sciatic nerves of tKO and control mice. Three different experiments of four to five animals per genotype are shown. Data were normalized to the IgG value as shown as relative enrichment. Unpaired *t*-test was used for comparations. (**D**) Alignment of the reads of the RNA-seq from three individual sciatic nerves of control and three tKO mice, both uninjured and at 20 days post crush (20 dpi). *Hdac9* gene is transcribed at detectable levels in the sciatic nerve of the uninjured tKO mice, whereas it is almost nondetectable in the control sciatic nerves. The tKO mice (but not the controls) increase additionally the expression of *Hdac9* gene during remyelination (20 dpi). (**E**) mRNA levels of *Hdac9* (as FPKMs) at 0, 1, 10, and 20 days post crush (dpi) in the RNA-seq experiment. Two-way analysis of variance (ANOVA) was used for statistical comparation. (**F**) *Mef2d* expression is increased early in development (P8) in tKO nerve (1.65 ± 0.18 in the tKO versus 0.97 ± 0.10 in controls; p = 0.025). RT-qPCR with mouse-specific primers for *Mef2d* was performed. The scatter plot, which include also the mean ± standard error (SE), shows the expression normalized to the housekeeping 18S. Four to five mice per genotype were used. Data were analyzed with the unpaired *t*-test with Welch's correlation. (**G**) mRNA levels of *Mef2d* (as FPKMs) at 0, 1, 10, and 20 days post crush (dpi) in the RNA-seq experiment. (**H**) A representative WB of protein extracts from tKO, control, and wild-type nerves at 10 dpi is shown. In the quantification, MEF2D protein was increased in the tKO nerves (2.31 ± 0.19 au in the tKO versus 1.33 ± 0.19 in controls; p < 0.0069). (**I**) Same for 20 dpi (2.19 ± 0.03 au in the tKO versus 1.33 ± 0.11 in controls; p < 0.0073). Densitometric analysis was done for three to four WB from the same number of mice and normalized to the control 20 dpi. Data were analyzed with the unpaired *t*-test. (**J**) MEF2D colocalizes with the transcription factor SOX10⁺, suggesting that it is expressed by Schwann cells. P60 sciatic nerves were fixed and submitted to immunofluorescence with the indicated antibodies. Nuclei were counterstained with Hoechst. Representative confocal images of sections obtained from the sciatic nerves of control and tKO mice are shown. Scale bar: 20 µm. (**K**) MEF2D binds to the *Hdac9* promoter in the tKO. ChIP-qPCR of 20 dpi nerves of tKO mice was performed using an anti-MEF2D-specific antibody. Five different experiments from four to five mice per genotype were performed. Data were analyzed with the unpaired *t*-test (*p < 0.05; **p < 0.01; ***p < 0.001; ns: no significant). See source data file one online (graphs source data) for more details.

The online version of this article includes the following figure supplement(s) for figure 9:

**Figure supplement 1.** *Hdac9* gene expression regulation.

Fluctuations in gene expression (noise) are a well-known phenomenon that has been described from bacteria to mammalian cells, and may have dramatic effects on fitness if they persist long enough (*Raser, 2010*). It has been suggested that some gene redundancies have been evolutionarily selected because they can reduce the harmful effects of gene expression noise (*Kafri et al., 2006*). Thus, the deleterious effect of the eventual decrease in the expression of a noisy gene pivotal for a determined biological process (such as differentiation) can theoretically be buffered by the expression of a redundant gene controlled by a different promoter.

Redundancies have been shown to be particularly relevant during development. One example is the couple *Myod/Myf-5*, which are master regulators of skeletal muscle development (*Sabourin and Rudnicki, 2000*). Similar to what happens with other redundant couples, *Myf-5* expression has a linear response that strictly dependents on the *Myod* gene expression dosage, and can likely contribute to reduce gene expression noise allowing skeletal muscle differentiation (*Kafri et al., 2006*).

It has been shown that slow oxidative fiber gene expression in skeletal muscles depends on gene redundancy between class IIa *Hdacs* (*Potthoff et al., 2007c*). Here, we show that the activation of the myelin gene expression program by Schwann cells is also ensured by *class IIa Hdacs* gene redundancy. Although the physiological role of genetic compensation within this family of proteins remains unknown, it is tempting to speculate that it could avoid potential fluctuations in class IIa *Hdac* gene dose ensuring Schwann cell differentiation and the proper myelination of the PNS.

But how is gene compensation regulated? Despite being documented many times in different organisms, our understanding of the underlying molecular mechanisms that control this process still remains limited (*El-Brolosy and Stainier, 2017*). Thus, genetic compensation of class I *Hdacs* has been previously described during myelin development (*Jacob et al., 2011*), although the mechanisms regulating this process have not been investigated. Here, we show that removal of *Hdac4* and *Hdac5* upregulates the compensatory overexpression of *Hdac7* in Schwann cells allowing, although with delay, myelin formation both during development and after nerve injury. Our data strongly suggest that this compensatory overexpression is regulated by the transcription factor JUN. In support of this tenet, we show that JUN binds to and induces the overexpression of *Hdac7* both in vitro and in vivo. Moreover, we also show *Hdac7* is not overexpressed in the nerves of dKO mice that lack *Jun* expression in Schwann cells. Interestingly, we found that HDAC4 binds to the promoter of *Hdac7* in differentiated Schwann cells, suggesting that other class IIa *Hdacs* contribute to maintain the expression of this gene at basal levels in normal nerves. In this scenario, the absence of *Hdac4* and *Hdac5*

in the dKO nerves might allow JUN to bind and stimulate the compensatory expression of *Hdac7* in Schwann cells (**Figure 10**).

We show that although *Hdac9* is normally not expressed by Schwann cells it is robustly upregulated in the nerves of the tKO. *Hdac9* is also upregulated, although much less, in the cKO7 and cKO4 mice. This support the idea that *Hdac9* gene is de novo expressed in response to the drop of class IIa *Hdacs* gene dose to allow myelination. Interestingly, *Hdac9* is also induced in the nerves dKO;Jun_cKO (**Figure 8G**), probably as a response to the drop of class IIa *Hdacs* in Schwann cells that cannot over-express *Hdac7* because of the absence of *Jun*.

To investigate the mechanisms that activate the expression of *Hdac9* in the tKO nerves we focused our attention in the MEF2 family of transcription factors, as they regulate *Hdac9* expression in other tissues (**Di Giorgio et al., 2020**; **Haberland et al., 2007**). Interestingly, we found MEF2D overex-pressed and bound to the *HDCA9* promoter in the tKO nerves.

It has been shown that MEF2 transcriptional activity is blocked by class IIa HDACs (**Haberland et al., 2007**). Thus, other class IIa HDACs can theoretically block the expression of *Hdac9* in control nerves. However, in the tKO, no class IIa *Hdac* is expressed in Schwann cells and a free of repression MEF2D might be able to induce *Hdac9* gene expression. Supporting this view, we found much more H3K9Ac associated with the *Hdac9* promoter in the tKO nerves.

Taken together, our data suggest that, the overexpressed and unrepressed promoter-bound MEF2D transcription factor, induces the de novo expression of *Hdac9* in the Schwann cells of the tKO mice (**Figure 10**).

Although adult tKO nerves have morphologically normal myelin, RNA-seq analysis showed 1270 genes differently expressed, the most robustly changed genes being upregulated. This suggests a predominantly gene repressive function for class IIa HADCs in Schwann cells, as is the case for other cell types (**Parra, 2015**; **Parra and Verdin, 2010**). The most robustly upregulated gene in these mice was *Tyrp1*, a gene involved in the stabilization of tyrosinase and the synthesis of melanin. Strikingly, we did not find expression of the tyrosinase gene in these nerves, suggesting a different role for *Tyrp1*. In fact, our data show that *Tyrp1* mRNA is not translated into protein (**Figure 2—figure supplement 1D**). It has been shown that *Tyrp1* mRNA indirectly promotes melanoma cell proliferation by sequestering miR-16 (**Gautron et al., 2021**; **Gilot et al., 2017**). Whether *Tyrp1* mRNA is also responsible for the increased cell proliferation of Schwann cells in the tKO nerve is something that needs to be clarified in the future.

It has been previously shown that a subgroup of melanocytes are formed from Schwann cell precursor cells (**Adameyko et al., 2009**). Because *Mpz-Cre* is already expressed by SCPs, our data points toward a role for class IIa HDACs in the repression of genes of the melanocytic lineage in these cells. Interestingly, it has been previously suggested that axonal derived signals repress SCP from going into the melanocytic lineage (**Graham, 2009**). Thus, our data suggest that these signals are not properly interpreted by tKO Schwann cells, precluding them to repress several melanocytic lineage genes. Notably, we found that melanocyte lineage genes are still expressed by the Remak Schwann cells of the adult tKO nerves. This misexpression could be in the origin of the alterations in the segregation of small size axons at the Remak bundles, a defect that remains during the whole life of the animal.

We also show that class IIa *Hdacs* removal delays remyelination after a nerve crush injury. Impor-tantly, tKO Schwann cells are efficiently reprogramed into the repair phenotype and myelin clearance is even more efficient than in control nerves, ruling out a problem in debris removal as the cause of remyelination delay. Although we do not know why myelin clearance is accelerated, it is worthy to mention that no increased autophagy markers neither macrophage numbers could be found in the tKO nerves. If myelin clearance is accelerated because changes in the rate axon degeneration or in ovoid formation is something that needs to be clarified in future experiments.

Genome-wide transcriptomic analysis of the injured tKO nerves showed that the number of differ-entially expressed genes increases after crush, and is maximum at 20 dpi. Many genes are robustly upregulated, particularly at 1 and 20 dpi, supporting further the idea that the main role of class IIa HDACs in Schwann cells is to repress gene expression. This agrees with the role of this family of deacetylases in other contexts (**Chang et al., 2004**; **Chang et al., 2006**; **Parra and Verdin, 2010**) where they work mainly as corepressors of transcription factors such as MEF2 and RUNX2 (**Bialek et al., 2004**; **Potthoff et al., 2007b**; **Vega et al., 2004**). Importantly, among the upregulated genes in

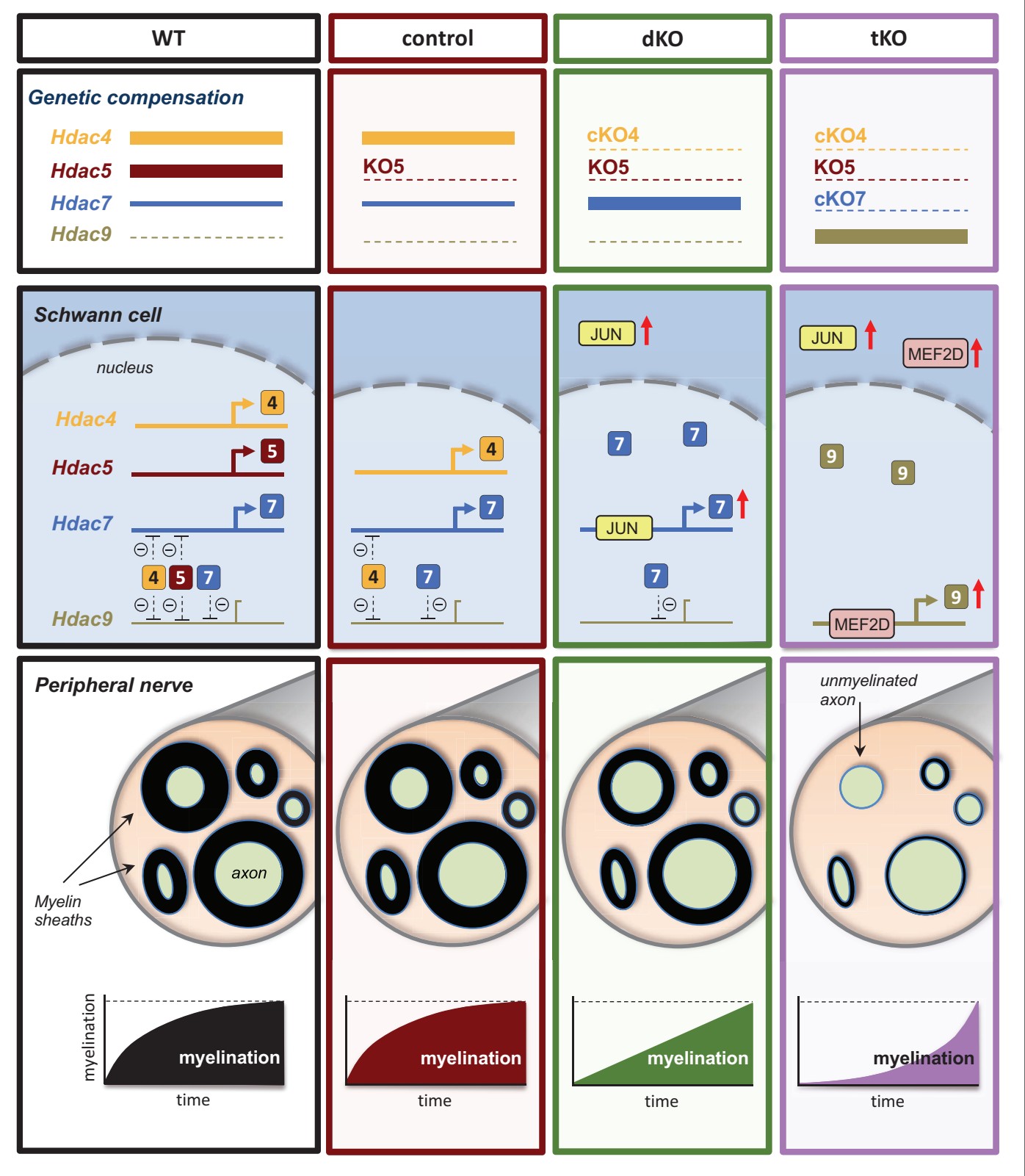

**Figure 10.** A graphical summary of the proposed model: In wild-type nerves (WT), the expression of *Hdac4*, *Hdac5*, and *Hdac7* allows myelin formation and blocks the expression of *Hdac9*. The removal of *Hdac5* (control) has no effects on myelination neither *Hdac9* expression. In the nerves of the dKO, JUN induces the compensatory overexpression of *HDac7*, allowing delayed myelination but having no effect on *Hdac9* gene expression. The simultaneous elimination of *Hdac4*, *Hdac5*, and *Hdac7* (tKO) induces the overexpression of *Mef2d*, which binds to the promoter and induce the compensatory expression of *Hdac9*, which after a long delay induces myelination.

the tKO injured nerves we found *Jun*, *Runx2*, *Gdnf*, *Ngfr*, and *Sox2*, all expressed by nonmyelinating and repair Schwann cells. We also found *Pou3f1* overexpressed in the tKO nerves. It has been shown that *Pou3f1* overexpression delays PNS myelination (*Ryu et al., 2007*). Thus, *Pou3f1* misexpression may also contribute to the delayed myelination of the tKO nerve both during development and after injury.

In a simplistic model, the failure of class IIa HDACs to downregulate *Jun* could indirectly induce the expression of other negative regulators of myelination controlled by this transcription factor. However, class IIa HDACs could also directly repress the expression of other negative regulators of myelination. To explore this idea, we performed a genome-wide mapping of genes that are direct targets of HDAC4. Importantly, we confirmed our previous results showing that HDAC4 binds to the promoters of *Jun*, *Runx2*, and *Gdnf* in Schwann cells (*Gomis-Coloma et al., 2018*). We found that HDAC4 also binds to the promoters of many other genes including other negative regulators of myelination such as *Sox2*, *Id2*, and *Hey2*. Interestingly, HDAC4 also binds to the promoter of *Pou3f1*. Thus, the direct repressive effect of class IIa HDACs is not circumscribed to *Jun* but is much wider, supporting the view that they work as a cAMP-regulated blocking hub for repressors of myelination.

Surprisingly, we found that HDAC4 is also bound to the promoter of *Mbp* and *Hmgcr*, two genes that are actively expressed during myelination. HDACs are usually bound to repressed genes and replaced by histone acetyl transferases (HATs) upon gene activation. However, it has been shown that class I HDACs are also bound, together with HAT, to the promoter regions of actively transcribed genes (*Wang et al., 2009*). Interestingly, the histones associated with these promoters are heavily acetylated, what makes these authors to propose that one function of HDACs is to remove the acetyl groups added by HATs in active genes to reset chromatin modification after gene activation. Although class IIa HDACs have no deacetylase activity, they recruit class I HDACs by forming a complex with NCOR1 and SMRT. Thus, the possibility exists that class IIa and class I HADCs have a similar role when bound to the promoter of highly active genes in myelinating Schwann cells.

In summary, the data presented in this manuscript unveil responsive backup circuits mediated by the transcription factors JUN and MEF2D, that coordinate genetic compensatory mechanisms within class IIa *Hdacs*, aimed at repressing the expression of negative regulators of myelination to ensure differentiation of Schwann cells in response to cAMP, and the generation of the myelin sheath during development and after nerve injury.

# Materials and methods
## Animal studies
All animal work was conducted according to European Union guidelines and with protocols approved by the Comité de Bioética y Bioseguridad del Instituto de Neurociencias de Alicante, Universidad Miguel Hernández de Elche and Consejo Superior de Investigaciones Científicas (http://in.umh-csic.es/). Reference number for the approved protocol was 2017/VSC/PEA/00022 tipo 2.

To avoid suffering, animals were sacrificed by cervical dislocation. The *Mpz-cre* mouse line is described in *Feltri et al., 1999*. *Mpz-cre*$^{-/-}$ littermates were used as controls. *Hdac4* floxed mice are described in *Lehmann et al., 2018* and *Potthoff and Olson, 2007a*. The *Hdac5* KO mouse line is described in *Chang et al., 2004*. *Hdac7* floxed mice are described in *Chang et al., 2006*. The Jun_OE and Jun_cKO mouse lines are described in *Fazal et al., 2017* and *Parkinson et al., 2008*. MGI ID can be found online (Key Resources Table). Experiments used mice of either sex on the C57BL/6 background.

## Plasmids
The luciferase reporter plasmid was generated by cloning the mouse *Hdac7* promoter into the NheI site of pGL4 Luciferase reporter plasmid (Promega). The mouse *Hdac7* promoter was amplified using Platinum SuperFi II DNA Polymerase (12361010, Thermo Fisher Scientific) and primers described online (Key Resources Table). The plasmid pCMV-Jun was a gift of Dr Marta Giralt (Universitat de Barcelona).

## Reporter activity assays

HEK293 cells were transfected with the indicated constructs and then lysed. Their luciferase activity was determined with the Luciferase Assay System (Promega) using the manufacturer's recommendations.

## Cell cultures

Schwann cells were cultured from sciatic nerves of neonatal rats as described previously (*Brockes et al., 1979*) with minor modifications. We used P3–P4 Wistar rat pups. The sciatic nerves were cut out from just below the dorsal root ganglia and at the knee area. During the extraction and cleaning, the nerves were introduced into a 35-mm cell culture dish containing 2 ml of cold Leibovitz's F-15 medium (Gibco) placed on ice. The nerves were cleaned, desheathed, and placed in a new 35-mm cell culture dish containing Dulbecco's Modified Eagle Medium (DMEM) with GlutaMAX and 4.5 g/l glucose (Gibco), with 1 mg/ml of collagenase A (Roche). Subsequently, they were cut into very small pieces using a scalpel and left in the incubator for 2 hr. Nerve pieces were homogenized using a 1-ml pipette, digestion reaction stopped with complete medium, and the homogenate poured through a 40 µm Falcon Cell Strainer (Thermo Fisher Scientific). We then centrifuged the homogenate at 210 × *g* for 10 min at room temperature and resuspended the pellet in complete medium supplemented with 10 µM of cytosine- β-D-arabinofuranoside (Sigma-Aldrich) to prevent fibroblast growth. The resuspended cells were then introduced into the poly-L-lysine-coated 35-mm cell culture dishes. After 72 hr, the medium was removed and cell cultures expanded in DMEM supplemented with 3% fetal bovine serum, 5 µM forskolin, and 10 ng/ml recombinant NRG1 (R&D Systems). Where indicated, cells were incubated in SATO medium (composed of a 1:1 mixture of DMEM and Ham's F12 medium [Gibco] supplemented with ITS [1:100; Gibco]), 0.1 mM putrescine, and 20 nM of progesterone (*Bottenstein and Sato, 1979*). HEK 293 cells were obtained from Sigma-Aldrich (Cat# 85120602). The cells were grown in noncoated flasks with DMEM GlutaMAX, 4.5 g/l glucose (Gibco) supplemented with 100 U/ml penicillin, 100 U/ml streptomycin, and 10% bovine fetal serum. Cells were transfected with plasmid DNA using Lipofectamine 2000 (Thermo Fisher Scientific) following the manufacturer's recommendations.

## Nerve injury

Mice were anesthetized with 2% isoflurane. To study nerve regeneration and remyelination (axonal regrowth inside the distal stump) we performed a nerve crush injury model. Briefly, the sciatic nerve was exposed at the sciatic notch and crushed three times for 15 s with three different rotation angles using angled forceps. To study the repair Schwann cell phenotype activation and myelin clearance, we performed a nerve cut to avoid nerve regeneration inside the distal stump. In this case, the sciatic nerve was exposed and cut at the sciatic notch using scissors. The wound was closed using veterinary autoclips (AutoClip System). The nerve distal to the cut or crush was excised for analysis at various time points after euthanasia. Contralateral uninjured sciatic nerves were used as controls. For Western blotting and mRNA extraction we used the first 8 mm of the distal stump of crush injured nerves, and 1 or 3 cm of the control nerve, respectively. For electron microscopy we used the first 5.5 mm of the distal stump of crush or cut injured nerves.

## Myelin clearance

Intact myelin sheaths were counted using transverse toluidine blue stained semithin sections (2 µm) of sciatic nerve at 5 mm from the nerve cut injury site. Whole nerve merged images were taken with a ×63 objective using a Leica Thunder Tissue Imager and quantified with ImageJ software.

## mRNA detection and quantification by RT-qPCR

Total mRNA from uninjured or injured sciatic nerves was extracted using used TRI reagent/chloroform (Sigma-Aldrich) and the mRNA was purified using a NucleoSpin RNA mini kit (Macherey-Nagel), following the manufacturer's recommendations. RNA quality and concentration were determined using a nanodrop 2000 machine (Thermo). Genomic DNA was removed by incubation with RNase free DNase I (Thermo Fisher Scientific), and 500 ng RNA was primed with random hexamers (Invitrogen) and retrotranscribed to cDNA with Super Script II Reverse transcriptase (Invitrogen). Control reactions were performed omitting retrotranscriptase. qPCR was performed using an Applied Biosystems QuantStudio 3 Real Time PCR System and 5× PyroTaq EvaGreen qPCR Mix Plus (CMB). To avoid

genomic amplification, PCR primers were designed to fall into separate exons flanking a large intron wherever possible. A list of the primers used can be found online (Key Resources Table). Reactions were performed in duplicates of three different dilutions, and threshold cycle values were normalized to the housekeeping gene 18S. The specificity of the products was determined by melting curve analysis. The ratio of the relative expression for each gene to 18S was calculated by using the $2^{-\Delta\Delta CT}$ formula. Amplicons were of similar size (≈100 bp) and melting points (≈85°C). Amplification efficiency for each product was confirmed by using duplicates of three dilutions for each sample.

## RNA sequencing analysis

Total RNA was isolated using the NucleoSpin RNA, Mini kit for RNA purification (Macherey-Nagel). The purified mRNA was fragmented and primed with random hexamers. Strand-specific first-strand cDNA was generated using reverse transcriptase in the presence of actinomycin D. The second cDNA strand was synthesized using dUTP in place of dTTP to mark the second strand. The resultant cDNA was then 'A-tailed' at the 3-end to prevent self-ligation and adapter dimerization. Truncated adaptors containing a T overhang were ligated to the A-tailed cDNA. Successfully ligated cDNA molecules were then enriched with limited cycle PCR (10–14 cycles). Libraries to be multiplexed in the same run were pooled in equimolar quantities. Samples were sequenced on the NextSeq 500 instrument (Illumina). Run data were demultiplexed and converted to fastq files using Illumina's bcl2fastq Conversion Software version 2.18 on BaseSpace. Fastq files were aligned to the reference genome (Mouse [GRCm38/Ensembl release 95] and analyzed with Artificial Intelligence RNA-SEQ [A.I.R.] software from Sequentia Biotech [https://www.sequentiabiotech.com/]).

## Antibodies

Immunofluorescence antibodies: JUN (Cell Signaling Technology, rabbit 1:800), Ki67 (Abcam, rabbit 1:100), L1 (Chemicon International, rat 1:50), MCAM (Origene, rabbit 1:200), MPZ (AvesLab, chicken 1:1000), NGFR (Thermo Fisher Scientific, mouse 1:100), SOX10 (R and D Systems, goat 1:100), donkey anti-goat IgG (H + L) Alexa Fluor 555 conjugate (Molecular Probes, 1:1000), donkey anti-rabbit IgG (H + L) Alexa Fluor 488 conjugate (Molecular Probes, 1:1000), donkey anti-chicken IgG (H + L) Alexa Fluor 488 conjugate (Jackson ImmunoResearch Labs, 1:1000), goat anti-rat IgG (H + L) Alexa Fluor 555 conjugate (Molecular Probes, 1:1000), Cy3 donkey anti-mouse IgG (H + L) (Jackson Immunoresearch, 1:500), and Cy3 donkey anti-rabbit IgG (H + L) (Jackson Immunoresearch, 1:500).

Antibodies used for Western blotting: CALNEXIN (Enzo Life Sciences, rabbit 1:1000), JUN (Cell Signaling Technology, rabbit 1:1000), GAPDH (Sigma-Aldrich, rabbit 1:5000), HDAC5 (Santa Cruz, mouse 1:500), KROX-20 (Millipore, rabbit 1:500), MCAM (Origene, rabbit 1:1000), MPZ (AvesLab, chicken 1:1000), NGFR (Covance, rabbit 1:1000), TYRP1 (Sigma-Aldrich, rabbit 1:1000), IgY anti-chicken HRP-linked (Sigma-Aldrich, 1:2000), IgG anti-mouse and IgG anti-rabbit HRP-linked (Cell Signaling Technology, 1:2000). A list of the antibodies used can be found online (Key Resources Table).

## Immunofluorescence

For immunofluorescence, mice were sacrificed by cervical dislocation and fresh frozen tissue was embedded in OCT (Sakura, 4583). Cryosections were cut at 10 µm on Superfrost Plus slides (Thermo Scientific, J1800AMNZ). Sections were thawed and fixed with 4% paraformaldehyde (PFA) for 5 min at room temperature. Then, samples were washed 3× in phosphate-buffered saline (PBS) 1× and immersed in 50% acetone, 100% acetone and 50% acetone for 2 min each. Then samples were washed 3× in PBS 1× and blocked in 5% donkey serum 0.1% bovine serum albumin (BSA) in PBS for 1 hr. Samples were incubated with the appropriate primary antibodies diluted in blocking solution overnight at room temperature. A list of the antibodies used can be found online (Key Resources Table). Samples were washed and incubated with secondary antibodies and DAPI in blocking solution for 1 hr at room temperature. Samples were mounted in Fluoromont G. Images were obtained at room temperature using a confocal ultraspectral microscope (Vertical Confocal Microscope Leica SPEII) with a ×63 Leica objective and using Leica LAS X software. Images were analyzed with ImageJ software.

## EM studies

Mice were sacrificed by cervical dislocation and sciatic nerve were exposed and fixed by adding fixative solution (2% PFA [15710, Electron Microscopy Sciences], 2.5% glutaraldehyde [16220, Electron

Microscopy Sciences]), 0.1 M cacodylate buffer, pH = 7.3 (12300, Electron Microscopy Sciences) for 15 min. Afterwards, the nerve was removed and placed in same fixative solution overnight at 4°C. Then, the nerve was washed in 0.1 M cacodylate buffer 3× for 15 min each. Then, the nerve was osmicated by adding 1% osmium tetroxide, 0.1 M cacodylate buffer, pH = 7.3 for one and half hour at 4°C. Then, the nerve was washed 2× with dd H₂O for 15 min each. Samples were dehydrated by washing progressively in: 25% ethanol for 5 min, 50% ethanol for 5 min, 70% ethanol for 5 min, 90% ethanol for 10 min, 100% ethanol for 10 min (×4), propylene oxide for 10 min (×3). They were then changed into a 50:50 mixture of Agar 100 resin:propylene oxide for 1 hr at RT. The final change was into a 75:25 mixture of Agar 100 resin:propylene for 2 hr at RT. Nerves were blocked in resin and left shaking O/N at RT. These nerves were re-blocked the following day with fresh resin for 2 hr at RT. The nerves were finally embedded in fresh resin and left in the oven for 24 hr at 65°C. Transverse ultrathin sections from nerves were taken 5 mm from the sciatic notch and mounted on film (no grid bars). Images were taken using a Jeol 1010 electron microscope with a Gatan camera analyzed with ImageJ software.

## Sodium dodecyl sulfate–polyacrylamide gel electrophoresis and immunoblotting

Sciatic nerves were homogenized at 4°C in RIPA buffer (PBS, 1% Nonidet P-40, 0.5% sodium deoxycholate, 0.1% sodium dodecyl sulfate SDS, and 5 mM EGTA) containing protease inhibitors (Mini Protease Inhibitor Cocktail; Sigma-Aldrich) and phosphatase inhibitors (Phosphatase Inhibitor Mini Tablets; Fisher Scientific). We homogenized the tissue using Bullet Blender Homogenizer BBX24-CE (Next Advance) and then sonicated for 4 min (30 s on/off) using a Biorruptor Pico (Diagenode). Protein concentrations were determined by the BCA method (Thermo Scientific). 10 µg of total protein was subjected to SDS–polyacrylamide gel electrophoresis (SDS–PAGE) and blotted on to Protran nitrocellulose membrane (Amersham Biosciences). Membranes were blocked using 5% milk (Sigma-Aldrich) in TBS 1% and incubated for 16 hr at 4°C with the indicated primary antibody, washed and incubated with secondary antibodies, and developed with ECL Prime (Amersham). Antibodies used can be found online (Key Resources Table). We used an Amersham Imager 680 machine (Amersham) for visualization. Measurements from the proteins of interest were normalized to loading control GAPDH and/or CALNEXIN. When normalized to both loading controls, a mean between the normalization with GAPDH and the normalization with CALNEXIN was used for analysis. The whole membrane Western blot images are shown in source data file four online.

## ChIP assays

ChIP: The ChIP assay was a modification of the method described by *Jang et al., 2006*. Schwann cell cultures were incubated in PBS/1% PFA for 10 min at room temperature, harvested by centrifugation (1000 × *g*, 5 min, 4°C) and washed with PBS. The pellet was resuspended in 1 ml of buffer A (50 mM HEPES–KOH, pH 8.1, 1 mM EDTA, 0.5 mM EGTA, 140 mM NaCl, 10% glycerol, 0.5% NP40, 0.25% Triton X-100, and protease inhibitors), homogenized, and sonicated (15 pulses of 30 s separated) in a Biorruptor Pico (diagenode). Chromatin was clarified by centrifugation at 17,000 × *g* for 3 min at room temperature. Protein concentration in the supernatant was quantified by the BCA method (Thermo Scientific). An aliquot was saved as input. The volume corresponding to 60–100 µg of protein was incubated with the corresponding antibody and Dynabeads Protein G (Life Technologies) overnight at 4°C to form immunocomplexes. For in vivo ChIP, freshly dissected uninjured and injured nerves were incubated in PBS/1% PFA for 10 min at room temperature and then quenched for 5 min with glycine 0.125 M. Nerves were washed in PBS for 10 min at 4°C and then lysed in 200 µl of buffer A, using Bullet Blender Homogenizer BBX24-CE (Next Advance). Nuclei were harvested by centrifugation (10,000 × *g*, 5 min, 4°C) and washed with 1 ml of buffer B (10 mM Tris-HCl, pH 8.0, 1 mM EDTA, 0.5 mM EGTA, 200 mM NaCl and protease inhibitors), and sonicated (15 pulses of 30 s separated) in a Biorruptor Pico (diagenode). Chromatin was clarified by centrifugation at 17,000 × *g* for 3 min at room temperature. Protein concentration in the supernatant was quantified by the BCA method (Thermo Scientific). AlAn aliquot was saved as input. The volume corresponding to 200–300 µg of protein was incubated with the corresponding antibody and Dynabeads Protein G (Life Technologies) overnight at 4°C to form immunocomplexes. In both cases, immune complexes were centrifuged (500 × *g*, 3 min) and washed twice with 1 ml of 'low-salt buffer' (0.1% SDS, 1% Triton X-100, 2 mM EDTA, 20 mM Tris, pH 8.1, 150 mM NaCl, and protease inhibitors; Roche), and then washed once with 1 ml of 'high-salt

buffer' (the same but with 500 mM NaCl) and washed three times with 1 ml of LiCl buffer (0.25 M LiCl, 1% IGEPAL, 1% sodium deoxycholate, 1 mM EDTA, 10 mM Tris, pH 8.1, and protease inhibitors). Chromatin from immunocomplexes and input was eluted with 200 µl of 1% SDS, 0.1 M NaHCO3, and 200 mM NaCl and incubated at 65°C for 6 hr (to break the DNA–protein complexes). DNA was purified using a column purification kit (GE Healthcare) and submitted to 5× PyroTaq EvaGreen qPCR Mix Plus (CMB) qPCR with the indicated primers.

The ChIP-Seq experiment was performed following a single-end sequencing strategy. High-quality reads were aligned against the reference genome (Rattus norvegicus (Rnor_6.0)) with Minimap2 (https://github.com/lh3/minimap2, version Minimap2-2.17 [r941]; RRID:SCR_018550). Read duplicates from the PCR amplification step in the sequencing process were removed with Picard tools (https://broadinstitute.github.io/picard/, version 2.23.8; RRID:SCR_006525) and only uniquely mapped reads were kept in the alignments. The uniquely mapped reads were obtained with SAMtools (http://www.htslib.org/) filtering by mapping quality ≥30. MACS2 (https://github.com/macs3-project/MACS, version 1.11; RRID:SCR_013291) was used for peak calling and the results were filtered by $-\log_{10}$ FDR > 3. To enable a more informative functional interpretation of experimental data, we identified genes close to or having ChIP-Seq tags on their sequence. ChIPseeker (http://bioconductor.org/packages/release/bioc/html/ChIPseeker.html) was used for this step. Rattus norvegicus (Rnor_6.0) was selected as annotation database (https://bioconductor.org/packages/release/data/annotation/html/TxDb.Rnorvegicus.UCSC.rn6.refGene.html).

## In vivo recording of CAP from mouse sciatic nerves

Mice were deeply anesthetized by intraperitoneal injection of 40 mg/kg ketamine and 30 mg/kg xylazine. The whole length of the right sciatic nerve was then exposed from its proximal projection into the L4 spinal cord to its distal branches innervating gastrocnemius muscles: tibial, sural, and common peroneal. Extracellular recording of CAPs was carried out by placing the proximal part of the sciatic nerve on an Ag/AgCl recording electrode with respect to a reference electrode (Ag/AgCl) placed inside the contralateral paw of the animal. For electrical stimulation, another electrode was placed in the distal part of the sciatic nerve just before its trifurcation. To avoid nerve desiccation and the consequent axonal death, the nerve was continuously lubricated with paraffin oil (Panreac). To selectively activate Aβ-, Aδ-, and C-fibers, we recorded CAPs evoked by graded electrical stimulations (5, 8, 10, and 15 V intensity, 0.03-ms pulse duration, Grass Instruments S88, A-M Systems) using both normal and inverse polarity. CAPs were amplified (×1000) and filtered (high pass 0.1 Hz, low pass 10 kHz) with an AC amplifier (DAM 50, World Precision Instruments) and digitalized and stored at 25 kHz in a computer using a CED micro-1401 interface and Spike2 v.7.01 software (both from Cambridge Electronic Design). In the CAPs, different waveform components corresponding to A- (β and δ) and C-fiber activation were easily distinguished by latency. The amplitude of the different components and the mean NCV were measured. The amplitude of each component was measured from the maximum negative to the maximum positive deflection (*Sdrulla et al., 2015*). For NCV measurement, we divided the distance between the stimulating and recording electrodes by the latency to the CAP component with the biggest amplitude (*Vleggeert-Lankamp et al., 2004*). The distance between the stimulating and recording electrodes was measured for each experiment using an 8/0 suture thread.

## Statistics

Values are given as means ± standard error (SE). Statistical significance was estimated with the Student's *t*-test with or without Welch's correction, one-way analysis of variance ANOVA with Tukey's multiple comparisons test, mixed ANOVA with Bonferroni's multiple comparisons test, chi-squared test and the Mann–Whitney *U*-test. A p value <0.05 was considered statistically significant. For the parametric tests (*t*-test and ANOVA), data distribution was assumed to be normal (Gaussian), but this was not formally tested. Analysis was performed using GraphPad software (version 6.0). Statistics for each experiment are described in more detail in the legends to figures.

## Acknowledgements

We would like to thank C Morenilla-Palao for advice in ChIP and other molecular biology experiments. We thank L Wrabetz and L Feltri for *Mpz-Cre* mice and E Olson for *Hdac* mice. We also thank P Morenilla-Ayala for technical assistance. We thank Prof Rhona Mirsky, University College London and

Shaline Fazal, University of Cambridge, for insightful comments on the manuscript. This work has been funded by grants from the Ministerio de Economía y Competitividad (BFU2016-75864R and PID2019-109762RB-I00), ISABIAL (UGP18-257 and UGP-2019–128) to H Cabedo, and Conselleria Educació Generalitat Valenciana (PROMETEO 2018/114) to J Gallar and H Cabedo. Predoctoral fellowships ACIF/2 017/169 from Generalitat Valenciana (to L Frutos-Rincón) and FPU16/00283from Ministerio de Universidades are also acknowledged. The Instituto de Neurociencias is a 'Center of Excellence Severo Ochoa' (Ministerio de Economía y Competitividad SEV-2013-0317). The authors declare no competing financial interests.

## Additional information

### Funding

| Funder | Grant reference number | Author |
|---|---|---|
| Ministerio de Economía y Competitividad | BFU2016-75864R | Hugo Cabedo |
| Ministerio de Economía y Competitividad | PID2019-109762RB-I00 | Hugo Cabedo |
| ISABIAL | UGP18-257 | Hugo Cabedo |
| ISABIAL | UGP-2019-128 | Hugo Cabedo |
| Conselleria de Cultura, Educación y Ciencia, Generalitat Valenciana | PROMETEO 2018/114 | Juana Gallar<br>Hugo Cabedo |
| Conselleria de Cultura, Educación y Ciencia, Generalitat Valenciana | ACIF/2 017/169 | Laura Frutos-Rincón |
| Ministerio de Educación, Cultura y Deporte | FPU16/00283 | Enrique Velasco |
| Wellcome Trust | 206634/Z/17/Z | Peter Arthur-Farraj |

The funders had no role in study design, data collection, and interpretation, or the decision to submit the work for publication.

### Author contributions

Sergio Velasco-Aviles, Nikiben Patel, Conceptualization, Investigation; Angeles Casillas-Bajo, Investigation, Project administration; Laura Frutos-Rincón, Enrique Velasco, Investigation; Juana Gallar, Funding acquisition, Supervision, Validation, Writing – original draft; Peter Arthur-Farraj, Resources, Supervision, Writing – review and editing; Jose A Gomez-Sanchez, Conceptualization, Formal analysis, Funding acquisition, Investigation, Methodology, Supervision, Validation, Writing – original draft, Writing – review and editing; Hugo Cabedo, Conceptualization, Formal analysis, Funding acquisition, Investigation, Methodology, Project administration, Supervision, Validation, Writing – original draft, Writing – review and editing

### Author ORCIDs

Sergio Velasco-Aviles http://orcid.org/0000-0002-9672-2264
Nikiben Patel http://orcid.org/0000-0002-0129-7622
Laura Frutos-Rincón http://orcid.org/0000-0002-0768-3735
Enrique Velasco http://orcid.org/0000-0001-7299-0750
Juana Gallar http://orcid.org/0000-0002-3559-3649
Peter Arthur-Farraj http://orcid.org/0000-0002-1239-9392
Jose A Gomez-Sanchez http://orcid.org/0000-0002-6746-1800
Hugo Cabedo http://orcid.org/0000-0002-1322-6290

### Ethics

All animal work was conducted according to European Union guidelines and with protocols approved by the Comité; de Bioética y Bioseguridad del Instituto de Neurociencias de Alicante, Universidad

Hernández de Elche and Consejo Superior de Investigaciones Científicas (http://in.umh.es/). Reference number for the approved protocol: 2017/VSC/PEA/00022 tipo 2.

### Decision letter and Author response
Decision letter https://doi.org/10.7554/eLife.72917.sa1
Author response https://doi.org/10.7554/eLife.72917.sa2

## Additional files

### Supplementary files
- Source data 1. Graphs source data.
- Source data 2. RNAseq source data.
- Source data 3. ChIPseq source data.
- Source data 4. Western blot source data.
- Reporting standard 1. ARRIVE guidelines checklist.

### Data availability
All data generated or analysed during this study are included in the manuscript and supporting file.

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

# Appendix 1

## Appendix 1—key resources table

| Reagent type (species) or resource | Designation | Source or reference | Identifiers | Additional information |
|---|---|---|---|---|
| Strain, strain background (*Mus musculus*, male and female) | cKO4 mouse *Mpz < Cre/+>;Hdac4< f/f>* | *Potthoff and Olson, 2007a* | Hdac4tm2.1Eno | C57BL/6 background, RRID:MGI:4418117 |
| Strain, strain background (*Mus musculus*, male and female) | KO5 mouse *Hdac5(-/-)* | *Chang et al., 2004* | Hdac5tm1Eno | C57BL/6 background, RRID:MGI:3056065 |
| Strain, strain background (*Mus musculus*, male and female) | cKO7 mouse *Mpz < Cre/+>;Hdac7< f/f>* | *Chang et al., 2006* | Hdac7tm2Eno | C57BL/6 background, RRID:MGI:3693628 |
| Strain, strain background (*Mus musculus*, male and female) | Jun_OE mouse *Mpz < Cre/+>;Rosa26< flox-stop-Jun/+>* | *Fazal et al., 2017* | Gt(ROSA)26Sortm15(Jun)Rsky | C57BL/6 background, RRID:MGI:6478892 |
| Strain, strain background (*Mus musculus*, male and female) | Jun_cKO mouse *Mpz < Cre/+>;Jun < f/f>* | Behrens et al. (2002) | *Juntm4Wag* | C57BL/6 background, RRID:MGI:2445420 |
| Strain, strain background (*Mus musculus*, male and female) | *Mpz < Cre/+>* | Jackson Laboratory | B6N.FVB-Tg(*Mpz-cre)26Mes/J* | *Mpz*-Cre mouse C57BL/6J background, RRID:IMSR_JAX:017927 |
| Antibody | anti- CANELXIN (rabbit polyclonal) | Enzo Life Sciences | Cat# ADI-SPA-860-D; RRID:AB_2038898 | WB (1:1000) |
| Antibody | anti- JUN (rabbit monoclonal) | Cell Signaling | Cat #9165; RRID:AB_2130165 | WB (1:1000), IF (1:800) |
| Antibody | anti- KROX-20 (rabbit polyclonal) | Millipore | Cat# ABE1374; RRID:AB_2715555 | WB (1:500) |
| Antibody | anti- Ki67 (rabbit polyclonal) | Abcam | Cat# ab15580; RRID:AB_443209 | IF (1:100) |
| Antibody | anti- F4/80 (rat monoclonal) | BioRad | Cat# MCA497GA; RRID:AB_323806 | IF (1:100) |
| Antibody | anti- GAPDH (rabbit polyclonal) | Sigma-Aldrich | Cat# G9545; RRID:AB_796208 | WB (1:5000) |
| Antibody | anti- HDAC4 (mouse monoclonal) | Sigma-Aldrich | Cat# H0163; RRID:AB_477042 | ChIP (10 mg) |
| Antibody | anti- HDAC5 (mouse monoclonal) | Santa Cruz | Cat# sc-133106; RRID:AB_2116793 | WB (1:1500) |
| Antibody | anti- IgG2a (mouse monoclonal) | Sigma-Aldrich | Cat# M7769; RRID:AB_1163540 | ChIP (10 mg) |
| Antibody | anti- L1 (rat monoclonal) | Chemicon International | Cat# MAB5272; RRID:AB_2133200 | IF (1:50) |
| Antibody | anti- LC3B (rabbit polyclonal) | Sigma-Aldrich | Cat# L7543; RRID:AB_796155 | WB (1:1000) |
| Antibody | anti- MCAM (rabbit monoclonal) | Origene | Cat# TA303592; RRID:AB_2143390 | WB (1:1000) IF (1:200) |
| Antibody | anti- MPZ (chicken polyclonal) | AvesLab | Cat# PZ0; RRID:AB_2313561 | WB (1:1000) IF (1:1000) |

*Appendix 1 Continued on next page*

*Appendix 1 Continued*

| Reagent type (species) or resource | Designation | Source or reference | Identifiers | Additional information |
|---|---|---|---|---|
| Antibody | anti- p75NTR (NGFR) (rabbit polyclonal) | Covance | Cat# PRB-602C; RRID:AB_291707 | WB (1:1000) |
| Antibody | anti- NGFR (mouse monoclonal) | Thermo Fisher Scientific | Cat# MA5-13314; RRID:AB_10982037 | IF (1:100) |
| Antibody | anti- SOX10 (goat polyclonal) | R&D Systems | Cat# AF2864; RRID:AB_442208 | IF (1:100) |
| Antibody | anti- TYRP1 (rabbit polyclonal) | Sigma-Aldrich | Cat# SAB2102617 RRID:AB_10611135 | WB (1:1000) |
| Antibody | anti- Rabbit IgG, HRP-linked (goat polyclonal) | Cell Signaling | Cat# 7074; RRID:AB_2099233 | WB (1:2000) |
| Antibody | anti- Mouse IgG, HRP-linked (horse polyclonal) | Cell Signaling | Cat# 7076; RRID:AB_330924 | WB (1:2000) |
| Antibody | anti- Chicken IgY (IgG) (rabbit polyclonal) | Sigma-Aldrich | Cat# A9046; RRID:AB_258432 | WB (1:2000) |
| Antibody | Cy3 anti-Rabbit IgG (H + L) (donkey polyclonal) | Jackson Immuno Research Labs | Cat# 711-165-152; RRID:AB_2307443 | IF (1:500) |
| Antibody | anti-Goat Alexa 555 Conjugated (donkey polyclonal) | Molecular Probes - Thermo Fisher | Cat# A21432; RRID:AB_2535853 | IF (1:1000) |
| Antibody | anti-Rabbit Alexa 488 Conjugated (donkey polyclonal) | Molecular Probes - Thermo Fisher | Cat# A21206; RRID:AB_2535792 | IF (1:1000) |
| Antibody | anti-Chicken Alexa 488 Conjugated (donkey polyclonal) | Jackson Immuno Research Labs | Cat# 703-545-155; RRID:AB_2340375 | IF (1:1000) |
| Antibody | anti-Rat Alexa 488 Conjugated (donkey polyclonal) | Molecular Probes - Thermo Fisher | Cat# A21208; RRID:AB_141709 | IF (1:1000) |
| Antibody | anti-Rat Alexa 555 Conjugated (goat polyclonal) | Molecular Probes - Thermo Fisher | Cat# A21434; RRID:AB_2535855 | IF (1:1000) |
| Antibody | Cy3 anti-Mouse IgG (H + L) (donkey polyclonal) | Jackson Immuno Research Labs | Cat# 715-165-151; RRID:AB_2315777 | IF (1:500) |
| Sequence-based reagent | *18 S_F* | *Gomez-Sanchez et al., 2009* | PCR primers | CGGCTACCACATCCAAGGAA |
| Sequence-based reagent | *18 S_R* | *Gomez-Sanchez et al., 2009* | PCR primers | GCTGGAATTACCGCGGCT |
| Sequence-based reagent | *Bdnf_F* | *Ma et al., 2016* | PCR primers | GGTATCCAAAGGCCAACTGA |
| Sequence-based reagent | *Bdnf_R* | *Ma et al., 2016* | PCR primers | GCAGCCTTCCTTGGTGTAAC |
| Sequence-based reagent | *Jun_F* | *Arthur-Farraj et al., 2012* | PCR primers | CCTTCTACGACGATGCCCTC |
| Sequence-based reagent | *Jun_R* | *Arthur-Farraj et al., 2012* | PCR primers | GGTTCAAGGTCATGCTCTGTTT |
| Sequence-based reagent | *Ednrb_F* | NM_007904.4 | PCR primers | CTGGCTCTGGGAGACCTACT |
| Sequence-based reagent | *Ednrb_R* | NM_007904.4 | PCR primers | GGGCACCAGCTTACACATCT |
| Sequence-based reagent | *Gdnf_F* | *Ma et al., 2016* | PCR primers | TCTCGAGCAGGTTCGAATGG |
| Sequence-based reagent | *Gdnf_R* | *Ma et al., 2016* | PCR primers | AAGAACCGTCGCAAACTTTACC |

*Appendix 1 Continued on next page*

*Appendix 1 Continued*

| Reagent type (species) or resource | Designation | Source or reference | Identifiers | Additional information |
|---|---|---|---|---|
| Sequence-based reagent | Hdac4_F | NM_207225.2 | PCR primers | GCGAGCACAGAGGTGAAGAT |
| Sequence-based reagent | Hdac4_R | NM_207225.2 | PCR primers | CGCTGGAAATGCAGTGGTTC |
| Sequence-based reagent | Hdac5_F | NM_001077696.1 | PCR primers | GGGGTGGAGGTGGAGGTAG |
| Sequence-based reagent | Hdac5_R | NM_001077696.1 (20) | PCR primers | CCGTAGCGCAGGGTCCAT |
| Sequence-based reagent | Hdac7_F | NM_001204275.1 | PCR primers | AGGAGCAAGAACTTCGGCAA |
| Sequence-based reagent | Hdac7_R | NM_001204275.1 | PCR primers | ACTGTTCTCTCAAGGGCTGC |
| Sequence-based reagent | Hdac9_F | NM_001271386.1 | PCR primers | CCCCTATGGGAGATGTTGAG |
| Sequence-based reagent | Hdac9_R | NM_001271386.1 | PCR primers | CAATGCATCAAATCCAGCAG |
| Sequence-based reagent | Hmgcr_F | *Gomez-Sanchez et al., 2009* | PCR primers | TGGATCGAAGGACGAGGAAAG |
| Sequence-based reagent | Hmgcr_R | *Gomez-Sanchez et al., 2009* | PCR primers | GAATTACGTCAACCATAGCTTCCG |
| Sequence-based reagent | Krox-20_F | NM_010118.3 | PCR primers | ACCCCTGGATCTCCCGTATC |
| Sequence-based reagent | Krox-20_R | NM_010118.3 | PCR primers | CAGGGTACTGTGGGTCAATGG |
| Sequence-based reagent | Mbp_F | *Gomez-Sanchez et al., 2009* | PCR primers | ATCCAAGTACCTGGCCACAG |
| Sequence-based reagent | Mbp_R | *Gomez-Sanchez et al., 2009* | PCR primers | CCTGTCACCGCTAAAGAAGC |
| Sequence-based reagent | Mcam_F | NM_023061.2 | PCR primers | GAAACGGCTACCCCATTCCT |
| Sequence-based reagent | Mcam_R | NM_023061.2 | PCR primers | AGCCACTGGACTCGACAATC |
| Sequence-based reagent | Mef2a_F | NM_001033713.2 | PCR primers | AGTAGCGGAGACTCGGAATTG |
| Sequence-based reagent | Mef2a_R | NM_001033713.2 | PCR primers | ATGCATCGTACACAGCTCCT |
| Sequence-based reagent | Mef2c_F | *Materna et al., 2019* | PCR primers | GTGCTGTGCGACTGTGAGAT |
| Sequence-based reagent | Mef2c_R | *Materna et al., 2019* | PCR primers | TCTGAGTTTGTCCGGCTCTC |
| Sequence-based reagent | Mef2d_F | NM_001310587.1 | PCR primers | GATCTGAACAATGCCCAGCG |
| Sequence-based reagent | Mef2d_R | NM_001310587.1 | PCR primers | GGCAGCTGGTAATCTGTGTTG |
| Sequence-based reagent | Mitf_F | NM_001113198.1 | PCR primers | GGAGCTCACAGCGTGTATTT |
| Sequence-based reagent | Mitf_R | NM_001113198.1 | PCR primers | TCCTTAATGCGGTCGTTTATGT |
| Sequence-based reagent | Mpz_F | *Gomez-Sanchez et al., 2009* | PCR primers | ACCAGACATAGT GGGCAAGACCTC |
| Sequence-based reagent | Mpz_R | *Gomez-Sanchez et al., 2009* | PCR primers | AAGAGCAACAGC AGCAACAGCACC |

*Appendix 1 Continued on next page*

*Appendix 1 Continued*

| Reagent type (species) or resource | Designation | Source or reference | Identifiers | Additional information |
| --- | --- | --- | --- | --- |
| Sequence-based reagent | *Ngfr*_F | *Fontana et al., 2012* | PCR primers | TGATGGAGTCGGGCTAATGTC |
| Sequence-based reagent | *Ngfr*_R | *Fontana et al., 2012* | PCR primers | AGATTCATCCCTCCACAAATGC |
| Sequence-based reagent | *Olig1*_F | *Ma et al., 2016* | PCR primers | AGCGATGTAGTTGCTTGGGAT |
| Sequence-based reagent | *Olig1*_R | *Ma et al., 2016* | PCR primers | CTGGCTCTAAACAGGTGGGAT |
| Sequence-based reagent | *Pou3f1*_F | NM_011141.2 | PCR primers | GAGCACTCGGACGAGGATG |
| Sequence-based reagent | *Pou3f1*_R | NM_011141.2 | PCR primers | TGATGCGTCGTTGCTTGAAC |
| Sequence-based reagent | *Prx*_F | NM_198048.2 | PCR primers | AGTGGCCAAGCTGAACATCC |
| Sequence-based reagent | *Prx*_R | NM_198048.2 | PCR primers | AGAACTCGACGTCAACAGGG |
| Sequence-based reagent | *Runx2*_F | NM_001146038.2 | PCR primers | GTCTTCCACACGGGGCAC |
| Sequence-based reagent | *Runx2*_R | NM_001146038.2 | PCR primers | GCCAGAGGCAGAAGTCAGAG |
| Sequence-based reagent | *Sox2*_F | *Quintes et al., 2016* | PCR primers | TCCAAAAACTAATCACAACAATCG |
| Sequence-based reagent | *Sox2*_R | *Quintes et al., 2016* | PCR primers | GAAGTGCAATTGGGATGAAAA |
| Sequence-based reagent | *Sox10*_F | NM_011437.1 | PCR primers | GAGCAAGCCGCACGTCAAGA |
| Sequence-based reagent | *Sox10*_R | NM_011437.1 | PCR primers | GTGGAGGTGAGGGTACTGGTC |
| Sequence-based reagent | *Shh*_F | *Ma et al., 2016* | PCR primers | CAGCGACTTCCTCACCTTCCT |
| Sequence-based reagent | *Shh*_R | *Ma et al., 2016* | PCR primers | AGCGTCTCGATCACGTAGAAGAC |
| Sequence-based reagent | *Tyrp1*_F | NM_031202.3 | PCR primers | CCGCTTTTCTCACATGGCAC |
| Sequence-based reagent | *Tyrp1*_R | NM_031202.3 | PCR primers | TCGCAGACGTTTTTCCCAGT |
| Sequence-based reagent | ChIP *Hdac7* Promoter_F | ID_84582 | PCR primers | CCCTCCACAATGACCCTCCTT |
| Sequence-based reagent | ChIP *Hdac7* Promoter_R | ID_84582 | PCR primers | GTGATCCGCTGTAATGCACTG |
| Sequence-based reagent | ChIP *Hdac9* Promoter_F | ID_687001 | PCR primers | GCTGCAATCACTCGGCCAT |
| Sequence-based reagent | ChIP *Hdac9* Promoter_R | ID_687001 | PCR primers | GCCCACAGGCACAGAAATAGA |
| Sequence-based reagent | ChIP *Pou3f1* Promoter_F | ID_192110 | PCR primers | CAGAAGGAGAAGCGCATGAC |
| Sequence-based reagent | ChIP *Pou3f1* Promoter_R | ID_192110 | PCR primers | CTCCCCAGGCGCATAAACG |
| Sequence-based reagent | *Jun*_FloxP_OE_F | *Fazal et al., 2017* | PCR primers | TGGCACAGCTTAAGCAGAAA |
| Sequence-based reagent | *Jun*_FloxP_OE_R | *Fazal et al., 2017* | PCR primers | GCAATATGGTGGAAAATAAC |

*Appendix 1 Continued on next page*

*Appendix 1 Continued*

| Reagent type (species) or resource | Designation | Source or reference | Identifiers | Additional information |
|---|---|---|---|---|
| Sequence-based reagent | Jun_FloxP_cKO_F | *Arthur-Farraj et al., 2012* | PCR primers | CCGCTAGCACTC ACGTTGGTAGGC |
| Sequence-based reagent | Jun_FloxP_cKO_F | *Arthur-Farraj et al., 2012* | PCR primers | CTCATACCAGTT CGCACAGGCGGC |
| Sequence-based reagent | Hdac4_FloxP_F | *Gomis-Coloma et al., 2018* | PCR primers | ATCTGCCCACCAGAGTATGTG |
| Sequence-based reagent | Hdac4_FloxP_R | *Gomis-Coloma et al., 2018* | PCR primers | CTTGTTGAGAAC AAACTCCTGCAGCT |
| Sequence-based reagent | Hdac5_FloxP_F | *Gomis-Coloma et al., 2018* | PCR primers | CAAGGCCTTGTG CATGCTGGGCTGG |
| Sequence-based reagent | Hdac5_FloxP_R | *Gomis-Coloma et al., 2018* | PCR primers | CTGCTCCCGTAG CGCAGGGTCCATG |
| Sequence-based reagent | Hdac5_FloxP_LacZ | *Gomis-Coloma et al., 2018* | PCR primers | GCCCGTTTGA GGGGACGACG ACAGTATTCG |
| Sequence-based reagent | Hdac7_FloxP_F | *Chang et al., 2006* | PCR primers | GTTGCAGGGTC AGCAGCGCAGGCTCTG |
| Sequence-based reagent | Hdac7_FloxP_R | *Chang et al., 2006* | PCR primers | CCAGTGGACGAG CATTCTGGAGAAAGG |
| Sequence-based reagent | Mpz-Cre_F | *Feltri et al., 1999* | PCR primers | CCACCACCTCT CTCCATTGCAC |
| Sequence-based reagent | Mpz-Cre_R | *Feltri et al., 1999* | PCR primers | GCTGGCCCAA ATGTTGCTGG |
| Commercial assay or kit | QIAquick PCR Purification Kit | Qiagen | Cat# 28,104 | |
| Commercial assay or kit | NucleoSpin RNA, Mini kit for RNA purification | Macherey-Nagel | Cat# 740955.50 | |
| Commercial assay or kit | Luciferase Assay System | Promega | Cat# E1500 | |
| Commercial assay or kit | Beta-Glo Assay System | Promega | Cat# E4720 | |
| Commercial assay or kit | Pierce BCA Protein Assay Kit | Thermo Scientific | Cat# 23,225 | |
| Commercial assay or kit | ECL Prime Western Blotting Detection Reagent | Amersham | Cat# RPN223 | Highly sensitive chemiluminescent detection reagent for Western blotting |
| Commercial assay or kit | ECL Western Blotting Analysis System | Amersham | Cat# RPN2109 | Chemiluminescent detection reagent for Western blotting |
| Commercial assay or kit | Invitrogen Dynabeads Protein G | Life Technologies | Cat# 10,004D | Superparamagnetic beads with recombinant Protein G covalently coupled to surface for Immunoprecipitation |
| Commercial assay or kit | MasterMix qPCR ROx PyroTaq EvaGreen 5 x | CMB | Cat# 08-24-00001 | |
| Chemical compound, drug | Agar 100 Resin | Agar Scientific | Cat# R1043 | Concentration: various, see methods |
| Chemical compound, drug | Dodecenyl Succinic Anhydride - DDSA | Agar Scientific | Cat# R1051 | Concentration: various, see methods |
| Chemical compound, drug | Methyl Nadic Anhydride - MNA | Agar Scientific | Cat# R1082 | Concentration: various, see methods |
| Chemical compound, drug | Benzyldimethylamine - BDMA | Agar Scientific | Cat# R1062 | Concentration: various, see methods |
| Chemical compound, drug | Paraformaldehyde 16% solution, EM grade | Electron Microscopy Sciences | Cat# 15,710 | Concentration: 2% |

*Appendix 1 Continued on next page*

*Appendix 1 Continued*

| Reagent type (species) or resource | Designation | Source or reference | Identifiers | Additional information |
|---|---|---|---|---|
| Chemical compound, drug | Glutaraldehyde 25% solution, EM grade | Electron Microscopy Sciences | Cat# 16,220 | Concentration: 2.5% |
| Chemical compound, drug | Sodium Cacodylate Trihydrate | Electron Microscopy Sciences | Cat# 12,300 | Concentration: 0.1 M |
| Chemical compound, drug | Ethanol absolute | J.T.Baker | Cat# 8025 | Concentration: various, see methods |
| Chemical compound, drug | Propylene Oxide, ACS Reagent | Electron Microscopy Sciences | Cat# 20,401 | Concentration: various, see methods |
| Chemical compound, drug | Leibovitz's L-15 Medium | Company | Cat# 11570396 | Concentration: various, see methods |
| Chemical compound, drug | DMEM GlutaMAX Medium | Gibco | Cat# 11574516 | Concentration: various, see methods |
| Chemical compound, drug | Forskolin | Gibco | Cat# F6886 | Concentration: various, see methods |
| Chemical compound, drug | rhNRG1-beta1 | Sigma-Aldrich | Cat# RYD-396-HB-050 | Concentration: various, see methods |
| Chemical compound, drug | FBS - Fetal Bovine Serum | R&D Systems | Cat# 11550356 | Concentration: various, see methods |
| Chemical compound, drug | Penicillin-Streptomycin | Fisher | Cat# 11548876 | Concentration: various, see methods |
| Chemical compound, drug | DMEM Ham's F12 Medium | Gibco | Cat# 11520396 | Concentration: various, see methods |
| Chemical compound, drug | Insulin-Transferrin-Selenium | Gibco | Cat# 41400-045 | Concentration: various, see methods |
| Chemical compound, drug | Putrescine | Gibco | Cat# P5780 | Concentration: various, see methods |
| Chemical compound, drug | Progesterone | Sigma-Aldrich | Cat# P0130 | Concentration: various, see methods |
| Chemical compound, drug | dbcAMP - Dibutyryl cAMP sodium salt | Sigma-Aldrich | Cat# D0627 | Concentration: various, see methods |
| Chemical compound, drug | LipoD293 DNA In Vitro Transfection Reagent | Sigma-Aldrich | Cat# SL100668 | Concentration: various, see methods |
| Chemical compound, drug | Mini Protease Inhibitor Cocktail | Roche | Cat# 11836153001 | Concentration: various, see methods |
| Chemical compound, drug | Phosphatase Inhibitor Mini Tablets | Fisher Scientific | Cat# 15691759 | Concentration: various, see methods |
| Chemical compound, drug | RNase A | Fisher Scientific | Cat# 10618703 | Concentration: various, see methods |
| Chemical compound, drug | Proteinasa K | Sigma-Aldrich | Cat# 3115836001 | Concentration: various, see methods |
| Chemical compound, drug | TRI Reagent | Sigma-Aldrich | Cat# T9424 | Concentration: various, see methods |
| Chemical compound, drug | Chloroform | Sigma-Aldrich | Cat# 319,988 | Concentration: various, see methods |
| Chemical compound, drug | DNaseI RNase Free | Thermo Fisher Scientific | Cat# EN0521 | Concentration: various, see methods |
| Chemical compound, drug | Deoxynucleotide Mix | Sigma-Aldrich | Cat# D7295 | Concentration: various, see methods |
| Chemical compound, drug | Random Primers | Invitrogen | Cat# 48190-011 | Concentration: various, see methods |
| Chemical compound, drug | DTT | Invitrogen | Cat# Y00147 | Concentration: various, see methods |

*Appendix 1 Continued on next page*

*Appendix 1 Continued*

| Reagent type (species) or resource | Designation | Source or reference | Identifiers | Additional information |
|---|---|---|---|---|
| Chemical compound, drug | RNaseOUT | Invitrogen | Cat# 10777-019 | Concentration: various, see methods |
| Chemical compound, drug | RevertAid Reverse Transcriptase | Thermo Fisher Scientific | Cat# EP0441 | Concentration: various, see methods |
| Software, algorithm | GraphPad Prism 9.0.0 | GraphPad Prism | RRID:SCR_002798 | |
| Software, algorithm | ImageJ | Open Source | RRID:SCR_003070 | |
| Software, algorithm | QuantStudio 3 Real-Time PCR Systems Software | Applied Biosystems | RRID:SCR_018712 | |
| Other | QuantStudio 3 Real-Time PCR Systems | Applied Biosystems | Cat# A28131 | Real Time PCR thermal cycler |
| Other | Biorruptor Pico sonication device | Diagenode | Cat# B01060010 | Sonication device |
| Other | Bullet Blender Homogenizer BBX24-CE | Next Advance | Cat# BBX24-CE | Tissue Homogenizer |
| Other | Zirconium Oxide Beads | Next Advance | Cat# ZrOB05 | For the homogenization of medium-tough tissue and cells |
| Other | Microplate Reader EZ Read 400 | Biochrom | Cat# 12694795 | 96-well plate reader for Pierce BCA protein assay kit |
| Other | Nitrocellulose Membrane | Amersham Hybond ECL | Cat# RPN203D | Protein blotting membrane (pore size: 0.45 mm) |
| Other | Amersham Imager 600 | Amersham | Cat# Amersham Imager 600 | Western blot (chemiluminescence) developer |
| Other | 4',6-diamidino-2-phenylindole (DAPI stain) | Thermo Fisher | Cat# D1306; RRID:AB_2629482 | Blue-fluorescent nucleic acid stain IF (1:1000) |

