## [Editor Report]

Analyzing Schwann cells which make myelin in the mammalian peripheral nervous system, the authors unravel how transcription factors can functionally substitute each other in the development of single/double/ and triple mutant mice. Functional redundancy and compensation is seen in many developmental systems, but has rarely been studied at that level of detail. The paper is thus of interest also to scientists beyond the field of glial cell biology.

---

## [Decision Letter]

**Decision letter after peer review:**

Thank you for submitting your article "A genetic compensatory mechanism regulated by c-Jun and Mef2d modulates the expression of distinct class IIa HDACs to ensure peripheral nerve myelination and repair" for consideration by *eLife*. We sincerely apologize for the delays in handling this manuscript. Your article has been reviewed by 2 peer reviewers (#2 and #3), and the evaluation has been overseen by a Reviewing Editor and Catherine Dulac as the Senior Editor. The reviewers have opted to remain anonymous.

The reviewers have discussed their reviews with one another, and the Reviewing Editor has drafted this to help you prepare a revised submission. As you can see, there was strong support for your paper. However, numerous constructive suggestions have been made how the manuscripts could be improved, none of which should be considered a major criticisms. Please see for details below.

*Reviewer #1 (Recommendations for the authors):*

The present manuscript provides new insight into the role of class II HDACs in peripheral nerve development and in repair. The data are in general of high quality, however there are several aspects that need to be addressed in order to improve the manuscript and to justify the central hypothesis of the authors:

Briefly

– The authors should carefully rewrite the manuscript, as the description of the single vs dKo vs triple KO is sometimes difficult to follow. This applies also to the timepoints that were analysed and, importantly, to the experimental techniques used in the nerve injury part. The injury type and analyzed time points are not very clear and structured or explained. Why do they use sometimes crush and sometimes cut?

– Some figures lack scale bars or the description of the scale bar. E.g. for the EM picture of the Remak fibers, the scale bar is not defined, so that axon size cannot be assessed.

– While it is clear that HDAC4/5/7 tKO impacts myelination and genetic regulation of SC development including Remak bundle architecture, the results do not quite justify the clear hypothesis of compensatory mechanisms. Remak bundle morphology should be addressed and discussed in the single mutants, cKO4 and cKO7 specifically. Discussion should also include HDAC9 expression in single mutants.

– Discussion of a physiological relevance of compensatory mechanisms between HDACs, i.e. why should loss of function of one/two/three HDACs occur in nature that would require compensation?

– Myelin clearance is not shown directly, but only indirectly by quantifying the number of still intact myelin profiles. Quantification of degeneration profiles and data on autophagy and/or macrophages would justify statements on myelin clearance.

– ChIPseq analysis was only performed for HDAC4, which shows convincing results, however only specific for HDAC4, thus, it is speculative to draw conclusions for all three HDACs.

– Is HDAC7 expression reduced in c-Jun cKO after injury?

– In general, the reviewer would recommend a more careful usage of the term "de novo" and "ectopic", with regard to the data presented.

*Reviewer #2 (Recommendations for the authors):*

The authors should consider reorganising the result section, clearly separating developmental and regenerative experiments. It would further be helpful for the reader to parse the text into clear paragraphs. Especially the result section would benefit from this.

It should be made clear how WB signals are normalised. Signals should be normalised to calnexin/GAPDH before comparisons are made between genotypes.

The lettering of some figures is too small. For example, in Figure 2 G (but also elsewhere) the gene names in the volcano plot are unreadable.

Discuss all Figure elements in the text. For example, Figure 2 C and D are not presented in the text.

---

## [Author Response]

Reviewer #1 (Recommendations for the authors):The present manuscript provides new insight into the role of class II HDACs in peripheral nerve development and in repair. The data are in general of high quality, however there are several aspects that need to be addressed in order to improve the manuscript and to justify the central hypothesis of the authors:Briefly– the authors should carefully rewrite the manuscript, as the description of the single vs dKo vs triple KO is sometimes difficult to follow. This applies also to the timepoints that were analysed and, importantly, to the experimental techniques used in the nerve injury part. The injury type and analyzed time points are not very clear and structured or explained. Why do they use sometimes crush and sometimes cut?

We apologize for not being clear enough. In the revised manuscript we have now rewritten the text to clarify the description of the different genotypes and phenotypes, as well as the analyzed time points. We have also included more information on the technique used for nerve injury. As we explained before, for most of the experiments we used the crush model of injury, that allow us to study the regeneration and remyelination of the nerve. We have used a cut model of injury only for the studies on myelin clearance and the activation of the Schwann cell repair phenotype. We apologize for not being clear enough and hope the reviewer will find it clearer in the revised version of the manuscript.

– Some figures lack scale bars or the description of the scale bar. E.g. for the EM picture of the Remak fibers, the scale bar is not defined, so that axon size cannot be assessed.

We apologize for these mistakes. We have now revised and fixed all the images with scale bars.

– While it is clear that HDAC4/5/7 tKO impacts myelination and genetic regulation of SC development including Remak bundle architecture, the results do not quite justify the clear hypothesis of compensatory mechanisms. Remak bundle morphology should be addressed and discussed in the single mutants, cKO4 and cKO7 specifically. Discussion should also include HDAC9 expression in single mutants.

After the suggestion of the reviewer, we have now quantified the Remak phenotype in the single and double KOs and compared with the tKO. We have also discussed the expression of HDAC9 in single mutants.

– Discussion of a physiological relevance of compensatory mechanisms between HDACs, i.e. why should loss of function of one/two/three HDACs occur in nature that would require compensation?

Several hypotheses have been proposed to explain the physiological relevance and mechanisms of genetic compensation. One of the hypotheses propose that it can reduce the harmful effects of gene expression noise. On this basis, in the revised manuscript we suggest and discuss more extensively that genetic compensation could avoid fluctuations in class IIa HDAC activity in Schwann cells allowing differentiation and the proper myelination of the peripheral nervous system.

– Myelin clearance is not shown directly, but only indirectly by quantifying the number of still intact myelin profiles. Quantification of degeneration profiles and data on autophagy and/or macrophages would justify statements on myelin clearance.

As explained before we have quantified both the number of intact myelin profiles and the expression of myelin protein zero (MPZ). To get further insight we have now also quantified autophagy by measuring LC3bI-II by WB and found no changes in the tKO. Also, no changes in other autophagy markers were found by RTqPCR. Thus, the consistently increased myelin clearance found in these mice is not caused by accelerated autophagy/ myelinophagy. These new data have been included in the revised manuscript. As indicated before we have also quantified the number of macrophages in the distal stumps at 4d after cut and found no changes between tKO and control.

– ChIPseq analysis was only performed for HDAC4, which shows convincing results, however only specific for HDAC4, thus, it is speculative to draw conclusions for all three HDACs.

We have now revised the text to avoid unnecessary generalizations.

– Is HDAC7 expression reduced in c-Jun cKO after injury?

As explained before to confirm the role of c-Jun in HDAC7 overexpression in the dKO nerves, we have generated the *dKO; c-Jun cKO* genotype and measured HDAC7 gene expression in the nerves of these mice. As is shown in Figure 8G of the revised manuscript, the absence of c-Jun in Schwann cells totally prevents the compensatory overexpression of HDAC7. Interestingly we have also found an increase in the expression of HDAC9 in these mice, probably to compensate their incapacity to upregulate HDAC7 (Figure 8G).

– In general, the reviewer would recommend a more careful usage of the term "de novo" and "ectopic", with regard to the data presented.

We have restricted the use of these terms only where it is justified in the revised manuscript.

Reviewer #2 (Recommendations for the authors):The authors should consider reorganising the result section, clearly separating developmental and regenerative experiments. It would further be helpful for the reader to parse the text into clear paragraphs. Especially the result section would benefit from this.

We now have tried to distinguish more clearly development and regenerative experiments in the revised version of the manuscript. However, we would like to point out that we follow, as much as possible, the timeline of the experiments we performed during the last years to obtain the necessary data for writing the manuscript. This, in our opinion, helps to make sense of the presented experiments and gives coherence to our approach. Unfortunately, it also makes very difficult to completely segregate the results of development from those of regeneration. For example, we weren’t aware of HDAC9 upregulation until we performed the RNAseq experiments of the regeneration of adult nerves. Then we had to come back to development to know when HDAC9 started to be upregulated. We apologize for these limitations and hope the reviewer will find enough the changes we have introduced in the revised version to clarify the manuscript.

It should be made clear how WB signals are normalised. Signals should be normalised to calnexin/GAPDH before comparisons are made between genotypes.

Practically all of the WB quantifications have been performed by normalizing to both calnexin and GADPH simultaneously. Only the WB of *Figure 2—figure supplement 1D* has been normalized uniquely to Calnexin because GAPDH is seen as double band in the iris. We apologize for not being clear enough. We have now put this information in the Material and methods section.

The lettering of some figures is too small. For example, in Figure 2 G (but also elsewhere) the gene names in the volcano plot are unreadable.

Size of lettering has been increased where necessary.

Discuss all Figure elements in the text. For example, Figure 2 C and D are not presented in the text.

We have revised the text and found all the figures discussed in the text, except Figure 2C and D. We discus both figures in the revised version of the manuscript.